# Cell surface-bound La protein regulates the cell fusion stage of osteoclastogenesis

Jarred M. Whitlock [1] ✉, Evgenia Leikina[1], Kamran Melikov[1], Luis Fernandez De Castro[2], Sandy Mattijssen [3], Richard J. Maraia[3], Michael T. Collins[2] & Leonid V. Chernomordik [1] ✉

Multinucleated osteoclasts, essential for skeletal remodeling in health and disease, are formed by the fusion of osteoclast precursors, where each fusion event raises their bone-resorbing activity. Here we show that the nuclear RNA chaperone, La protein has an additional function as an osteoclast fusion regulator. Monocyte-to-osteoclast differentiation starts with a drastic decrease in La levels. As fusion begins, La reappears as a low molecular weight species at the osteoclast surface, where it promotes fusion. La's role in promoting osteoclast fusion is independent of canonical La-RNA interactions and involves direct interactions between La and Annexin A5, which anchors La to transiently exposed phosphatidylserine at the surface of fusing osteoclasts. Disappearance of cell-surface La, and the return of full length La to the nuclei of mature, multinucleated osteoclasts, acts as an off switch of their fusion activity. Targeting surface La in a novel explant model of fibrous dysplasia inhibits excessive osteoclast formation characteristic of this disease, highlighting La's potential as a therapeutic target.

Bone-resorbing osteoclasts are responsible for essential, life-long skeletal remodeling, and their dysfunction is a major contributor to bone diseases affecting >200 million individuals worldwide[1], including osteoporosis, fibrous dysplasia (FD), Paget's disease and osteopetrosis[2–6]. Multinucleated osteoclasts are formed by the successive fusion of mononucleated precursor cells[7]. The number of nuclei per syncytial osteoclast, thus, the number of fusion events that generated each cell, directly correlates with the cell's ability to resorb bone[8–10]. Moreover, the number and size of osteoclasts are significantly altered in many bone diseases[11,12]. Recent studies suggest that during their relatively long lifetime[13] osteoclasts can go through additional rounds of cell fusion. Following their initial formation, multinucleated osteoclasts can undergo fission producing smaller daughter cells, termed osteomorphs, that can then migrate and fuse again to form mature multinucleated osteoclasts in a different location[14]. Despite the fundamental role of cell-cell fusion in osteoclast formation and bone remodeling, the mechanisms underpinning this process as well as other cell-cell fusion processes in normal physiology and in disease[15–17] remain to be fully understood. A number of proteins, including DC-STAMP, OC-STAMP, syncytin 1, annexin A5 (Anx A5), S100A4, CD47 and SNX10[18–23], have been implicated in osteoclast fusion, however, how osteoclasts regulate their fusion and arrive at the "right size" to fulfil their biological function remains elusive.

Osteoclasts derive from monocytes when stimulated by macrophage colony-stimulating factor (M-CSF), receptor activator of NF-kappaB ligand (RANKL), and other cytokines released by bone-forming osteoblasts and osteocytes[24]. In vitro, M-CSF and RANKL together are sufficient to elicit osteoclastogenesis. First, M-CSF stimulates the generation of adherent mononucleated osteoclast precursors. Second, RANKL commits these precursors to osteoclastogenesis and fusion[25]. While exploring proteomic changes during this stepwise process, we unexpectedly discovered that osteoclastogenesis involves lupus La

[1]Section on Membrane Biology, Eunice Kennedy Shriver National Institute of Child Health and Human Development, National Institutes of Health, Bethesda, MD 20892, USA. [2]Skeletal Disorders and Mineral Homeostasis Section, National Institute of Dental and Craniofacial Research, National Institutes of Health, Bethesda, MD 20892, USA. [3]Section on Molecular and Cell Biology, Eunice Kennedy Shriver National Institute of Child Health and Human Development, National Institutes of Health, Bethesda, MD 20892, USA. ✉e-mail: jarred.whitlock@nih.gov; chernoml@mail.nih.gov

protein (*SSB* gene product). La, also referred to as LARP3 and La autoantigen, is generally recognized as an abundant and ubiquitous RNA-binding protein[26]. La has a nuclear localization sequence (NLS) at its C-terminus in addition to other intracellular trafficking signals[27] that result in La being observed almost exclusively in the nucleus of human cells[28]. The best-characterized function of nuclear La is to protect precursor tRNAs from exonuclease digestion through specific interactions between La's highly conserved, N-terminal La domain and the 3' ends of tRNA. In addition to its nuclear functions, La shuttles to the cytoplasm[29] and assists in the correct folding of some mRNAs, acting as an RNA chaperone[30]. In a few specialized biological processes (e.g., apoptosis, viral infection, serum starvation), La protein is non-phosphorylated at phospo-Ser-366, loses its NLS via proteolytic cleavage, and this low molecular weight (LMW) species traffics to the surface of the cells[27,31–34]. However, the biological function of this cleaved, surface La, if any, is unknown.

Here, we report that osteoclast formation is accompanied by and depends on drastic changes in the steady-state level, molecular species, and intracellular localization of La protein. We demonstrate that human and murine La functions as a regulator of osteoclast fusion and impacts osteoclasts' ability to resorb bone. Surprisingly, La, present in primary human monocytes, nearly disappears in M-CSF-derived osteoclast precursors. RANKL-induced commitment to osteoclastogenesis drives the reappearance of La protein at the surface of committed, fusing osteoclasts. As osteoclast fusion plateaus, LMW La disappears and higher molecular weight, phosphorylated, full-length protein (FL-La) is observed within the nuclei of mature, multinucleated osteoclasts. Perturbing La expression, cleavage or surface function inhibits osteoclast fusion, while exogenous, surface La promotes fusion. Moreover, the mechanism by which La promotes osteoclast fusion is independent of La's ability to interact with RNA through its highly conserved La domain. Indeed, a C-terminal portion of La, lacking the La domain and RNA recognition motif 1 (RRM1) is sufficient to promote fusion between human osteoclasts. Our findings indicate that, while La protein plays an ancient, well-described and essential role in the RNA biology of all eukaryotes, La has been adapted in mammals to also serve as an osteoclast fusion manager. In this highly specific role on the surface of fusing osteoclasts, La may present a promising target for the treatment of bone diseases stemming from perturbed bone turnover.

## Results

### Formation of multinucleated osteoclasts involves La protein

Human osteoclastogenesis was modeled by treating primary monocytes with M-CSF to derive mononucleated osteoclast precursors to which recombinant RANKL was subsequently added to obtain multinucleated osteoclasts that readily resorb bone[18] (Fig. 1a, b, Fig. S1a–c). Osteoclast precursors begin fusing at ~2 days following RANKL addition and after ~5 days reach sizes (~ 5–10 nuclei/cell) characteristic of mature multinucleated osteoclasts[10,35,36] (Fig. 1b, Fig. S1c).

While evaluating changes in expression of some proteins associated with osteoclastogenesis, we serendipitously discovered a distinct protein that was nearly absent in M-CSF-derived precursors but abundantly expressed in osteoclasts following ~3 days of RANKL stimulated osteoclastogenesis, when the cells were rapidly fusing (Fig. 1c, arrow). Using mass spectrometry analysis, we identified this protein as La (Fig. S1d). The low level of La in M-CSF-derived macrophage precursors of osteoclasts was unexpected, as La is generally considered an abundant, ubiquitous protein[26,37–40].

Western blot analysis with mouse α-La antibody (α-La mAb) confirmed that La is highly expressed in monocytes, markedly reduced in M-CSF-derived osteoclast precursors and returned to high steady-state levels during RANKL-induced osteoclast formation (Fig. 1d). Our data suggest that La's tight regulation during osteoclastogenesis is likely carried out at the protein level, as M-CSF derived precursors

contain even more La transcript (gene *SSB*) than after RANKL application (Fig. S1e). When La returns in RANKL derived, fusogenic osteoclasts, it appears as two distinct, temporally separated molecular species (Fig. 1e). A low molecular weight (LMW La) species is detected at timepoints correlated with osteoclast fusion (Fig. 1e vs. 1b) and is replaced by a higher molecular weight species, corresponding to full-length La (FL La), as fusion slows and osteoclasts reach a mature size.

In addition to changes in molecular weight, osteoclastogenic differentiation of human monocytes is accompanied by a dramatic change in La's location within cells. Canonically, La exhibits robust nuclear staining[26], as illustrated for HeLa cells stained with α-La mAb (Fig. S1f). In contrast, M-CSF-derived osteoclast precursors exhibit minimal La staining. Addition of RANKL produced abundant La signal in committed, fusing osteoclasts, however, in contrast to other human cell types and tissues[26,27], La appeared as distinct, non-nuclear puncta throughout osteoclasts during early stages of osteoclast fusion (day 3 post-RANKL) (Fig. 1f). This dramatic increase in α-La mAb La staining in fusing osteoclasts is consistent with our biochemical and mass-spectrometry data (Fig. 1d, Fig. S1d).

While during timepoints of robust fusion (day 2–3) we observed La largely in the osteoclast cytoplasm (Fig. 1f), as fusion approached a plateau (days 4, 5), La gradually shifted to a predominantly nuclear localization in osteoclasts (Fig. 1g). We found that this change in localization of La correlates with its phosphorylation status. Most La in human cells is phosphorylated at Ser 366 and retained within the nucleus[41]. While La is abundantly expressed in osteoclasts both at timepoints of active fusion and when fusion plateaus (Fig. 1e, g), osteoclasts exhibited minimal to no signal when stained with antibodies specific for La phosphorylated at Ser366 (α-p366 La rAb) at timepoints associated with robust fusion (day 3, Fig. S1g). As fusion plateaued, we readily observed significant p366 La staining, but this staining was only observed in the nuclei (day 5, Fig. S1g). Finding that osteoclast fusion is accompanied not only by the appearance of a LMW species of La (Fig. 1e) but also by an absence of phosphorylation at Ser366 is consistent with a previous report demonstrating that dephosphorylation of La Ser366 is prerequisite for La cleavage-producing LMW La[33]. Staining patterns with α-La mAb and α-p366 La rAb at different time points taken together with our biochemical observations suggest that cytoplasmic La in fusing osteoclasts largely corresponds to a non-phosphorylated, LMW species, and that the observed plateau in osteoclast fusion is associated with a loss of LMW, cytoplasmic La and a shift to FL, nuclear La.

Cytosolic localization of La at the time of fusion was also observed during osteoclastogenic differentiation and fusion of RAW 264.7 derived murine osteoclast precursors (Fig. 2a). Furthermore, Western blot analysis of cell lysates collected separately from mostly mono-nucleated cells and from mostly multinucleated cells (see Methods) indicated that the robust fusion at day 3 post-RANKL was accompanied by a drastic increase in steady-state levels of La (Fig. 2b). These findings suggested that La dependence in osteoclast formation is conserved in humans and mice. Note that in distinction to human cells, in the case of RAW 264.7 cells, following the stage of active fusion by day 5, the levels of La returned to lower prefusion levels.

Finding that osteoclastogenic differentiation is accompanied by drastic changes in the expression and localization of La motivated us to explore whether La is functionally involved in osteoclast formation. We found that RNAi-mediated reduction of La transcript (*SSB*) (Fig. 2c) drastically inhibits human osteoclast fusion (Fig. 2d, e). To identify the functional form of La associated with osteoclastogenesis, we focused on the relationship between the appearance of LMW La and osteoclast fusion. Earlier reports demonstrate that during apoptotic progression human La is cleaved by caspases at Glu-375, removing La's NLS[33,42]. We found that overexpression of La 1-375, mimicking this cleaved species, greatly promoted fusion in both RAW 264.7 derived, murine

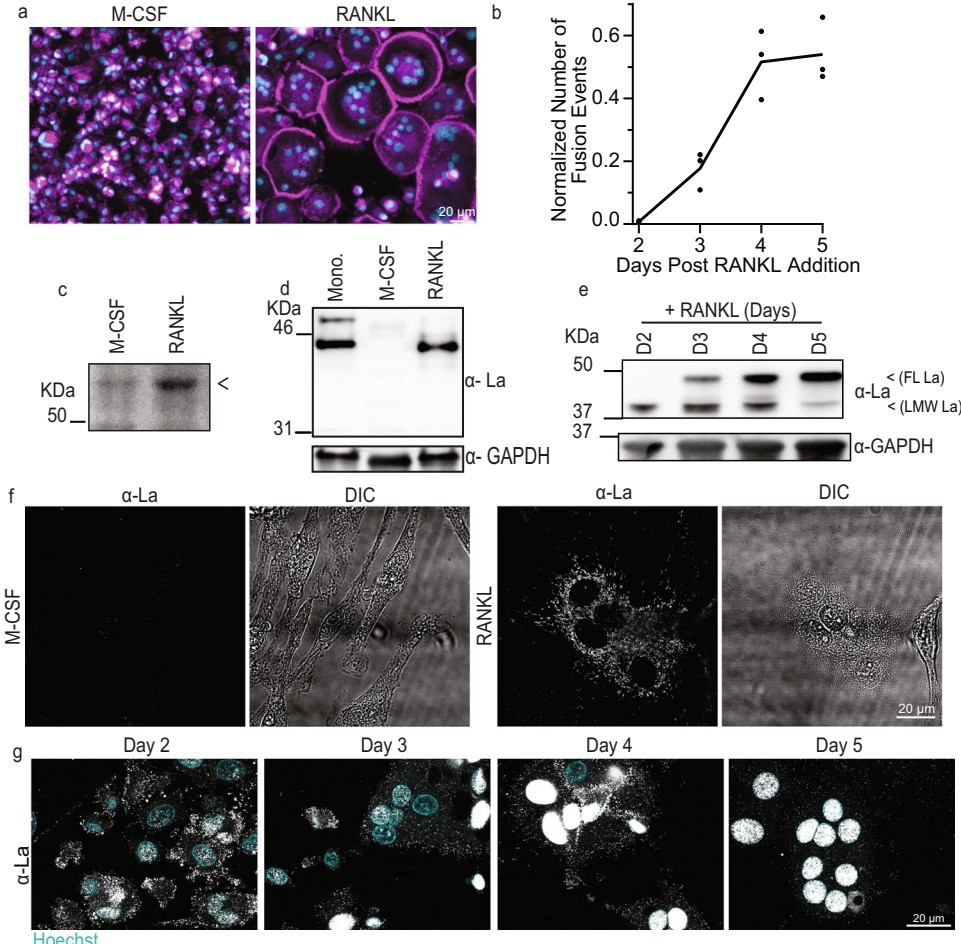

**Fig. 1 | Osteoclastogenic differentiation is accompanied by drastic changes in the steady-state levels and localization of La molecular species.**
**a** Representative images of stages of osteoclastogenic derivation of human monocytes after M-CSF (6 days, referred to as "M-CSF") and after M-CSF (6 days) followed by M-CSF + RANKL (5 days, "RANKL"), respectively. (Magenta = Phalloidin-Alexa488, Cyan = Hoechst). **b** Quantification of the number of fusion events normalized to the total number of nuclei observed over time following RANKL addition. (*n* = 3) Each point represents an average of >7500 nuclei scored.
**c** Representative Bis-Tris PAGE separation and silver staining of whole protein lysates from M-CSF derived osteoclast precursors and at 3 days post RANKL application. Lysates were ran until the 50 KDa marker nearly ran off the 4–12% polyacrylamide gel to achieve maximal separation of proteins at this molecular

weight, leading to the band of interest appearing misleadingly heavy. <denotes band of interest excised from both lanes and evaluated via mass spectrometry.
**d** Representative Tris-Glycine Western blot with α-La mAb evaluating La expression in whole protein lysates from primary human monocytes and the osteoclastogenic stages depicted in a. **e** Representative Tris-Glycine Western blot with α-La mAb evaluating the time course of La expression following RANKL addition. (α-GAPDH loading control). **f** Representative immunofluorescence images of La in M-CSF derived osteoclast precursors and at 3 days post RANKL application (α-La mAb).
**g** Representative immunofluorescence images of La in forming osteoclasts 2–5 days post RANKL application (α-La mAb). Cells were stained for La at the described timepoints with membrane permeabilization. Source data are provided as a Source Data file.

osteoclasts and monocyte derived, human osteoclasts (Fig. 2f–i). In contrast, overexpression of an uncleavable mutant of FL La, D371A,D374A La (point mutations disrupting La's predicted caspase cleavage sites) in human osteoclasts had no effect on their fusion despite similar expression levels, suggesting that formation of multinucleated osteoclasts depends on LMW La (Fig. 2h, i and Fig. S2a, b). Further supporting this point, we found that the pan-caspase inhibitor z-VAD and a specific inhibitor of caspase 3, z-DEVD, blocked the production of LMW La in differentiating osteoclasts (Fig. S3a, b). Blocking the caspase-dependent production of LMW La resulted in the retention of La within the nuclei of unfused osteoclasts (Fig. S3c) and significantly perturbed the ability of osteoclasts to form multinucleated syncytia (Fig. S3d, see also[43]). Taken together our data suggest that formation of multinucleated osteoclasts involves caspase 3-cleaved, non-phosphorylated LMW La.

To summarize, osteoclastogenesis is accompanied by drastic changes in La levels, molecular species, and location within fusing osteoclasts. A cleaved, non-nuclear La species promotes osteoclast

formation, and as cells arrive at a mature size, LMW La is replaced by FL La detected in the nuclei of syncytial osteoclasts.

## Cell surface-associated La regulates the cell fusion stage of osteoclast formation

In our characterization of La's role in osteoclastogenesis, we first explored whether La exerted its function in the formation of osteoclasts indirectly by altering the expression of factors implicated in osteoclastogenic differentiation or osteoclast fusion. While La expression in diverse cell types influences the steady-state levels of many transcripts/proteins[39], RNAi suppression of La did not alter the steady-state transcript levels of the essential osteoclastogenesis factors NFATc1 and CTSK or transcripts coding for the fusion-associated proteins syncytin 1, Anx A5, S100A4 or the lipid scramblase TMEM16F (Fig. S4a)[18]. Moreover, RNAi suppression of La did not alter the steady-state levels of several proteins previously linked to osteoclast fusion, including syncytin 1, Anx A5, TMEM16f or DC-STAMP (Fig. S4b). Therefore, while La knockdown inhibited the formation of osteoclast

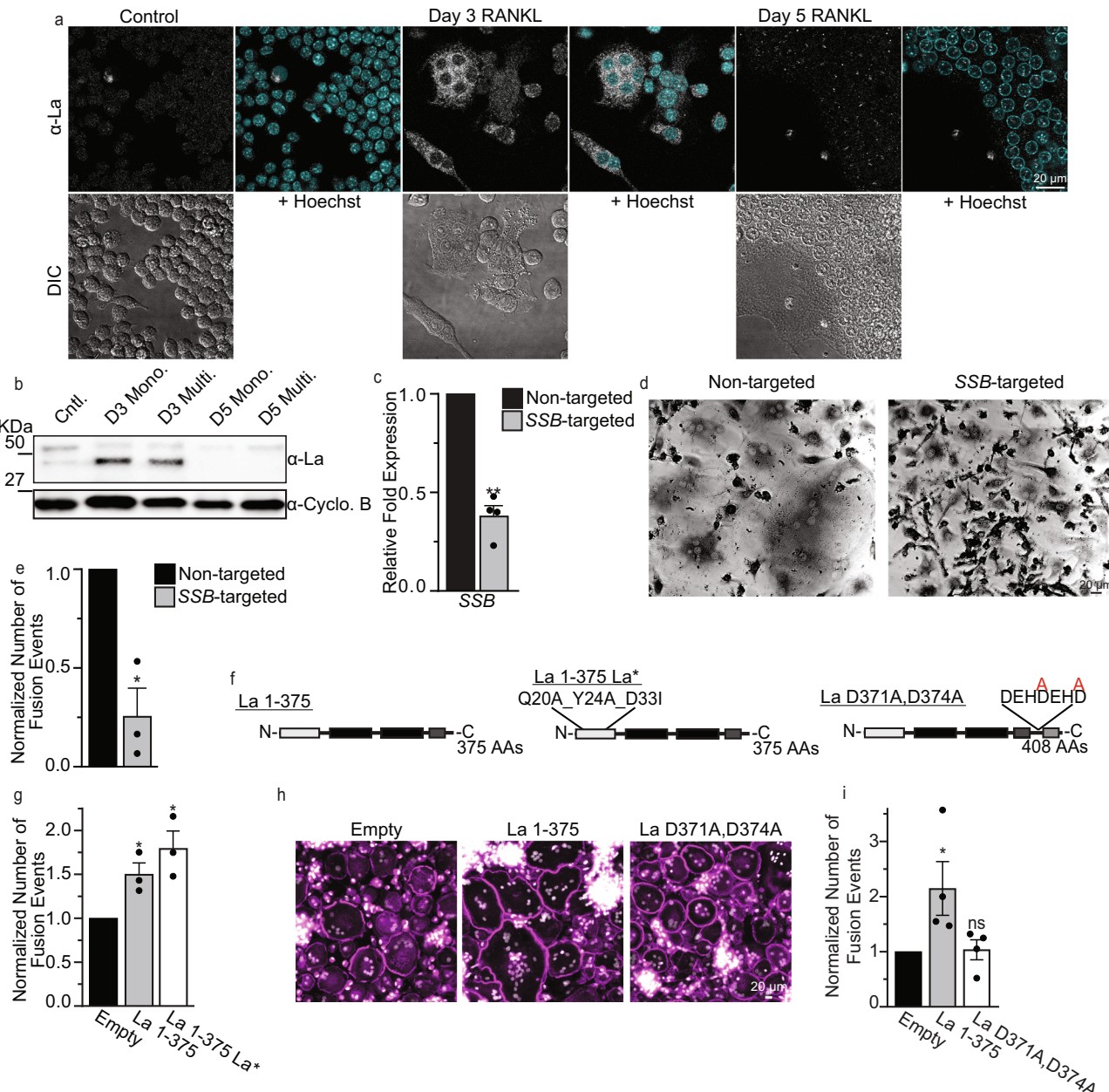

**Fig. 2 | Osteoclast formation depends on truncated La, but the function of the La domain is dispensable. a** Representative immunofluorescence images of La in RAW 264.7 prior to mRANKL addition (Cntl.), 3 days post mRANKL addition and 5 days post mRANKL addition when we routinely observe massive, multinucleated osteoclasts like the one imaged here. (α-La mAb). **b** Representative tris-glycine Western blot of whole cell lysates taken from murine, RAW 264.7 treated as in **a** (α-La rAb). mRANKL-treated cells were enriched into mononucleated (Mono.) or multinucleated (Multi.) populations as described in the Methods. (Cyclophilin B (α-Cyclo B) loading control). **c** qPCR evaluation of *SSB* in human osteoclast precursors treated with siRNA at day 1 post-RANKL addition. ($n = 4$) ($P = 0.0043$). **d** Representative, phase contrast images of non-targeted and *SSB*-targeted human osteoclasts, as in **c**, stained for TRAP. **e** Quantification of the number of fusion events in formation of syncytia with 3+ nuclei in the experiments like the one in **d**.

Fusion was scored at day 3. ($n = 3$) ($P = 0.0306$). **f** Topological illustrations of LMW La 1-375, "uncleavable" La D371A,D374A and "La*" (=La 1-375 Q20A_Y24A_D33I). **g** Quantification of the number of fusion events in syncytia with 3+ nuclei in RAW 264.7 cells transfected with empty, La 1-375 or La 1-375 Q20A_Y24A_D33I expression plasmids. ($n = 3$) ($P = 0.0213$ and $0.0173$, respectively). **h** Representative fluorescence images of human monocyte-derived osteoclasts transfected with empty, La 1-375 or La D371A,D374A expression plasmids. (Magenta = Phalloidin-Alexa488, Grey = Hoechst). **i** Quantification of the number of fusion events in syncytia with 3+ nuclei in **h**. ($n = 4$) ($P = 0.0205$ and $0.325$, respectively). **c, e, g, i** Statistical significance was evaluated via one-tailed paired t-tests. * = $P < 0.05$. ** = $P < 0.001$. Data are presented as mean values +/- SEM. Source data are provided as a Source Data file.

syncytia, it did not grossly impact osteoclast differentiation or machinery critical for cell-cell fusion. To further explore the mechanism by which La influenced the formation of osteoclasts but not their differentiation, we assessed whether the formation of multinucleated osteoclasts depends on La's well-characterized RNA binding function. This highly conserved function is largely based on high-affinity interactions between the La domain and its high-affinity oligo(U)-3' binding site common to RNA polymerase III transcripts. To assess the requirement of high-affinity interactions between the La domain and transcripts in osteoclastogenesis, we overexpressed a mutant La 1-375 with three-point mutations known to functionally impair La domain function, Q20A/Y24A/D33I[44,45] (La 1-375 La*). We found that La 1-375 La*

promoted formation of multinucleated osteoclasts as robustly as wild-type La 1-375, indicating that the La domain's high-affinity for RNA polymerase III transcripts is dispensable for La's role in osteoclast formation (Fig. 2f, g).

In addition to La's ability to promote osteoclast formation in the absence of a functional La domain, the hypothesis that La's role in the formation of multinucleated osteoclasts is separate from its canonical role in RNA metabolism was supported by findings suggesting the importance of La membrane association. As noted above, in differentiating osteoclasts La loses its NLS and appears in punctate structures throughout the cell. We enriched proteins from RANKL-committed osteoclasts at timepoints when cells were actively fusing into soluble, cytosolic, or membrane-associated protein fractions. As expected, we found actin mostly in the cytosolic fraction, transmembrane RANK receptor in the membrane fraction, and the peripheral membrane protein Anx A5 in both fractions (Fig. 3a). While La is putatively considered a soluble protein, in differentiating osteoclasts, we found La in both cytosolic and membrane-associated fractions,

suggesting that La was unexpectedly associating with membranes during osteoclast formation (Fig. 3a).

In earlier reports, La cleavage in apoptotic cells was associated with the detection of La on the cell surface[33,42], however, whether this surface La plays some cellular function or operates simply as an antigen remains unknown. To assess whether osteoclast La traffics to the cell surface following cleavage, we stained fusing osteoclasts with α-La mAb under non-permeabilizing conditions (Fig. 3b, c). In contrast to the osteoclast peripheral membrane protein Fish, which is enriched during osteoclast fusion and binds to the cytoplasmic leaflet of the plasma membrane (PM)[46], La abundantly decorated the surface of fusing human osteoclasts (Fig. 3b). We found that suppressing La steady-state levels with RNAi significantly reduced this La surface staining, further confirming the specificity of this staining (Fig. S4c, d). Moreover, this surface pool of La is not exclusive to human osteoclasts. We also observed significant La surface staining in RAW 264.7 derived, murine osteoclasts (Fig. 3c), suggesting surface La is a feature common to fusing osteoclasts in mammals. Using surface staining of human

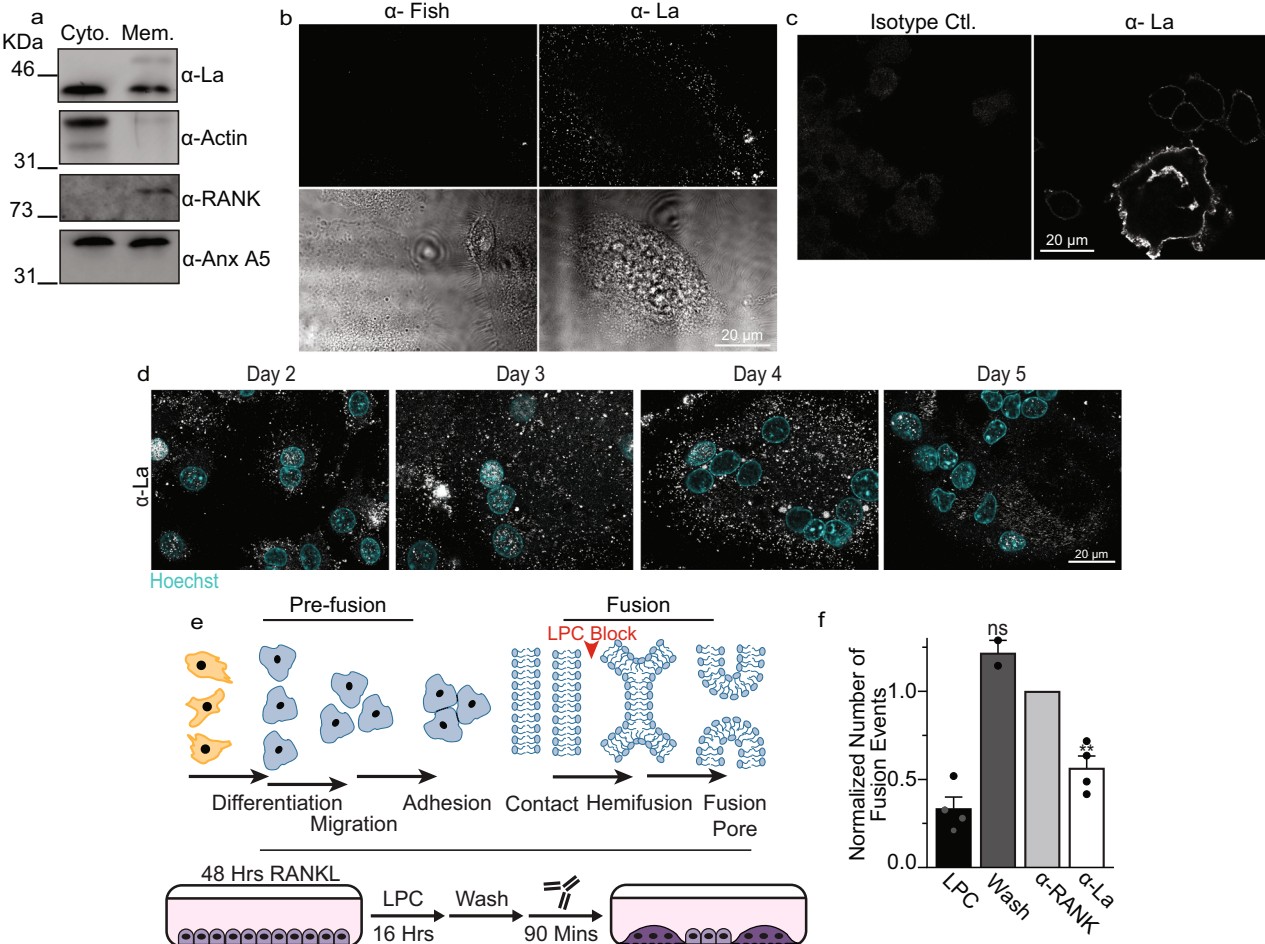

**Fig. 3 | La associates with membranes, traffics to the surface and controls osteoclast membrane fusion. a** Westerns of cytosolic vs membrane associated protein fractions from human osteoclasts. **b** Representative immunofluorescence images comparing surface staining with α-Fish/TKS5 antibody or α-La mAb in human osteoclasts under non-permeabilized conditions (top) and DIC (bottom). **c** Representative immunofluorescence images comparing surface staining of isotype control or α-La mAb in RAW 264.7 derived osteoclasts under non-permeabilized conditions. (α-La mAb). **d** Representative immunofluorescence images of cell surface La in forming human osteoclasts 2–5 days post RANKL application. Cells were stained with α-La mAb at the described timepoints without membrane permeabilization. **e** Cartoons illustrating the stepwise process of the formation of multinucleated osteoclasts (top), and our approach for isolating

membrane fusion stage from the preceding stages of osteoclast differentiation (bottom). Application of the hemifusion inhibitor LPC following 48 h of RANKL elicited osteoclastogenesis allows pre-fusion differentiation stages but blocks hemifusion, synchronizing cells. Removing LPC allows us to specifically probe membrane fusion between osteoclasts. **f** Quantification of human osteoclast fusion decoupled from pre-fusion stages and synchronized as depicted in **e** with fusion in the presence of 5 μg/ml α-La mAb and with no antibodies added (Wash) normalized to 5 μg/ml α-RANK control. (wash $n = 2$; others $n = 4$) ($P = 0.0086$ and 0.1330, respectively). "LPC" – fusion observed without removal of LPC. Statistical significance was assessed via one-tailed paired t-test. * = $P < 0.05$. ** = $P < 0.001$. Data are presented as mean values +/- SEM. Source data are provided as a Source Data file.

osteoclasts with α-La mAb at different days post RANKL application, we observed the transient increase in surface La at the time points associated with robust fusion (Fig. 3d vs. Figure 1b).

We then assessed whether La at the surface of human osteoclasts functions at the cell fusion stage of osteoclastogenesis. All cell-cell fusion events in development and tissue maintenance proceed through slow (days), asynchronous differentiation processes that prepare fusion-competent cells[15]. Then, fusion of plasma membranes occurs by the rapid (minutes) progression from the initial formation of hemifusion connections to fusion pores that unite the volumes of two cells (Fig. 3e, top). We decouple these steps in the formation of multinucleated syncytia using the hemifusion inhibitor lysophosphatidylcholine (LPC)[18]. LPC's inverted cone shape is not conducive to the concave geometry of the hemifusion stalk, so ready-to-fuse cells are trapped upstream of hemifusion. After removing LPC, cells undergo synchronized fusion relatively rapidly (within 90 mins), affording us the ability to assess the function of proteins specifically in the membrane fusion stage of osteoclast formation decoupled from upstream differentiation processes (Fig. 3e, bottom). We accumulated ready-to-fuse, RANKL committed cells in the presence of LPC, and then lifted this hemifusion blockade by washing out LPC (Fig. 3f). Application of α-La mAbs at the time of LPC removal significantly inhibited synchronized osteoclast membrane fusion (Fig. 3f). In contrast, isotype-matched antibodies targeting the plasma membrane receptor RANK at the surface of osteoclasts had no effect despite comparable levels of binding between α-La mAb and α-RANK antibody to the surface of synchronized cells following LPC removal (Fig. S5a). While RANKL-RANK signaling triggers upstream osteoclastogenic differentiation, inhibition of RANK following hemifusion synchronization fails to inhibit membrane fusion, as fusion itself does not depend on the activity of RANK[18]. As additional support, we found that α-La rabbit

antibody (α-La rAb), in contrast to isotype control IgG, also blocks synchronized osteoclast fusion (Fig. S5b). α-p366 La Ab did not suppress fusion, likely because La phosphorylated at Ser366 does not contribute to the fusion stage of osteoclast formation (Fig. S5b). Importantly, thanks to the strong, non-specific binding of any rabbit IgG to the abundant Fc receptors on the surface of human macrophage-lineage cells[47], the levels of osteoclast surface binding by α-La rAb, α-p366 La rAb and control IgG are similar (Fig. S5c). Thus, our finding that only α-La rAb inhibits fusion cannot be explained by non-specific steric hindrance of cell surface-associated immunoglobulins.

In contrast to the fusion-inhibiting effects of antibodies targeting surface La, application of recombinant La dramatically promoted osteoclast fusion. Application of FL La (La 1-408), truncated La (La 1-375) or truncated, RNA binding mutant La 1-375 La* outside fusing osteoclasts significantly promoted the formation of multinucleated syncytia (Fig. 4a–c). This promotion was not observed when recombinant La was heat inactivated. Recombinant La 1-375 La* promoted fusion similarly to La 1-408 and La 1-375, confirming that La's high-affinity interactions with RNA polymerase III transcripts are not required for La's role in regulating osteoclast fusion (Fig. 4a–c). Moreover, the ability of FL La to promote osteoclast fusion demonstrates that FL La itself is not fusion incompetent and suggests that proteolytic processing and/or dephosphorylation of La are important for fusion because of their role in the delivery of La to the cell surface.

To further resolve the contributions of La domains critical for RNA binding, La and RRM1 domains[26,27,44,45], we split La 1-375 into La 1-187 and La 188-375. We found that La 188-375 greatly promoted the formation of multinucleated osteoclasts, whereas La 1-187 had no effect (Fig. 4d). These data demonstrate that the La domain, RRM1 and La's C-terminal 33 AAs are dispensable for La's role in osteoclast formation (Fig. 4d). Importantly, La promotes the formation of

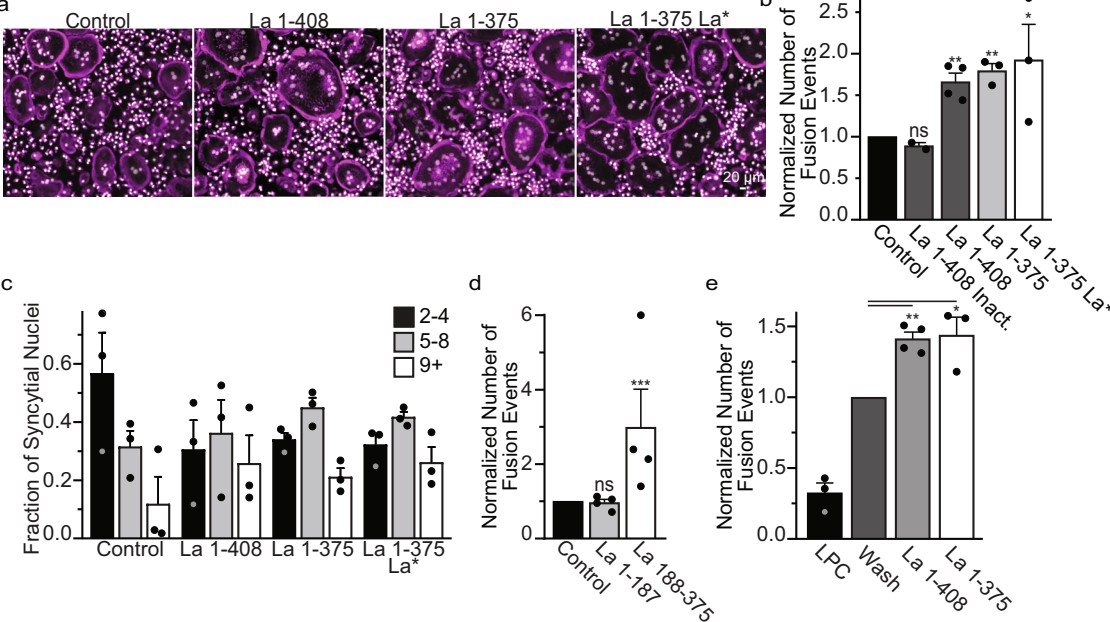

**Fig. 4 | Recombinant La promotes osteoclast fusion. (a)** Representative fluorescence images of human osteoclasts 3 days post RANKL addition without or with the overnight (end of day 2 post RANKL) addition of recombinant heat-inactivated La 1-408, La 1-408, La 1-375 or La 1-375 Q20A/Y24A/D33I. Recombinant proteins were added at ~40 nM at the end day 2 post-RANKL addition, and cells were fixed the next morning. (Magenta = Phalloidin-Alexa488, Grey = Hoechst)
**(b)** Quantification of **a**. (inactivated n = 2; La 1-408 n = 4, others n = 3,) (P = 0.1232, 0.0015, 0.0035 and 0.0491, respectively) **(c)** Quantification of the fraction of nuclei in fused cells that were present in syncytia of various sizes from **a**. (n = 3).
**(d)** Quantification of the number of fusion events with or without the addition of La

1-187 or La 188-375. Recombinant proteins were added at ~40 nM at the end day 2 post-RANKL addition, and cells were fixed the next morning. (n = 4) (P = 0.36 and 0.0002) **(e)** The quantification of synchronized fusion events (as illustrated in Fig. 3d) without (wash) and with addition of recombinant La species. "LPC" – indicates that the hemifusion inhibitor was left until fixation. (LPC and La 1–375 n = 3; Wash and La 1-408 n = 4) (P = 0.001 and 0.03, respectively.) **(b–e)** Statistical significance was evaluated via one-tailed paired t-test. In (**b, d, e**) the data were normalized to those in control (no protein added in **b, d**, and wash with no proteins added **e**). Data are presented as mean values +/- SEM. Source data are provided as a Source Data file.

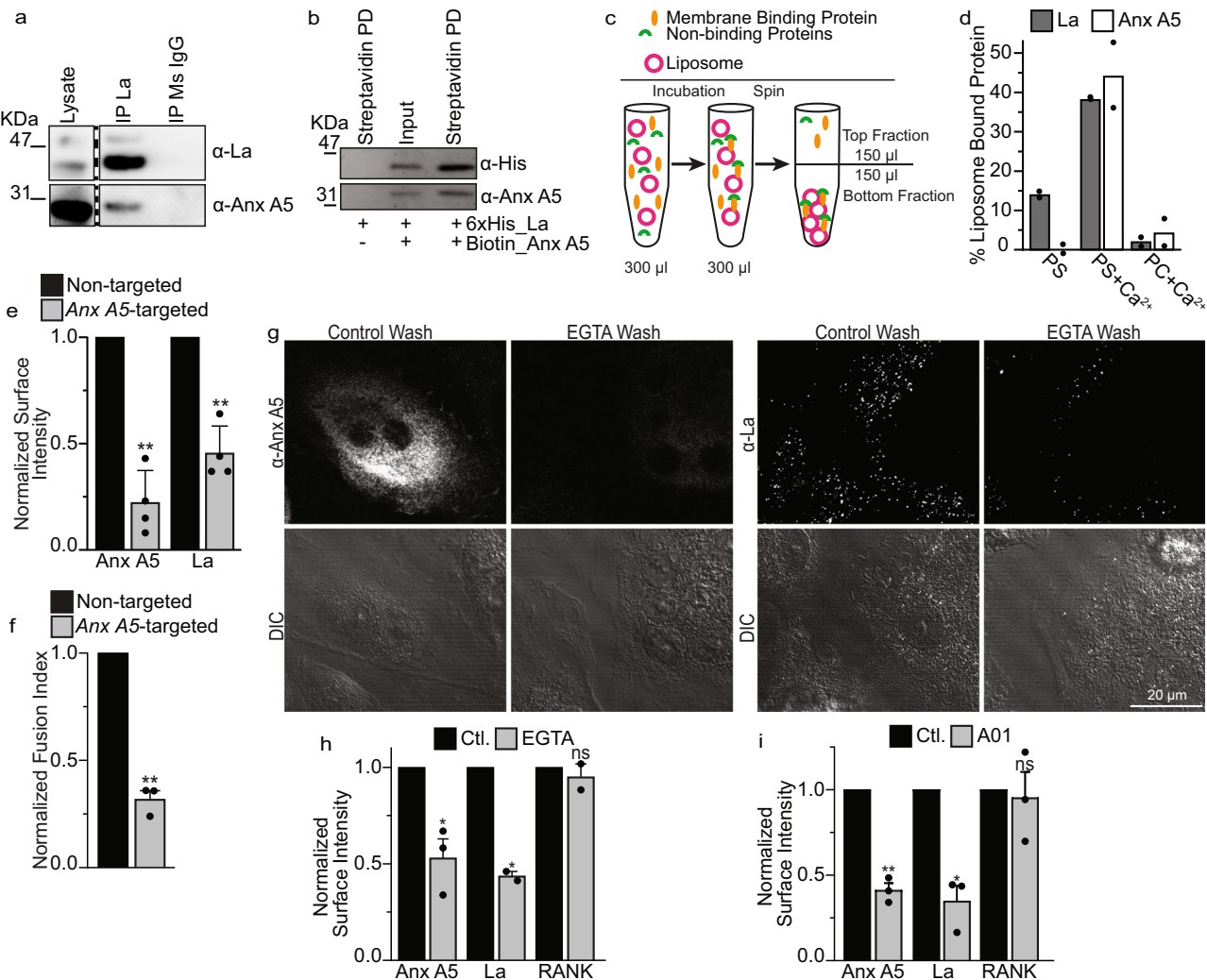

**Fig. 5 | La associates with the surface of osteoclasts by direct interactions with Anx A5. a** Immunoprecipitation of osteoclast lysates 3 days post-RANKL addition. La supermolecular complexes were captured on immunomagnetic beads via α-La mAb or isotype control and complexes were blotted with rabbit antibodies raised towards the targets of interest (α-La rAb for La). Lanes from the same blot are presented at the same intensity. Lanes of interest were cropped and placed beside one another, divided by a dashed line. **(b)** Representative Western blot of magnetic, streptavidin pull-down of Biotin-Anx A5. Lane 2 = La+Anx A5 input before pull-down and Lane 1 = La alone and Lane 3 = La+Anx A5 after pull-down. **c** A cartoon illustration of our approach to identify membrane affinity by comparing protein contents in the Bottom fraction containing, along with soluble proteins, liposome-bound proteins and in the Top fraction containing soluble proteins and depleted of liposome-bound proteins. **(d)** Quantification of the enrichment of recombinant La and Anx A5 in the bottom fraction containing pelleted liposomes (n = 2).

**(e)** Quantification of surface fluorescence intensity of Anx A5 or La following either non-targeted or Anx A5-targeted siRNA in human osteoclasts (n = 4) (P = 0.009 and 0.004, respectively). **(f)** Quantification of fusion events from **e** (n = 3) (P = 0.007). **(g)** Representative immunofluorescence images of Anx A5 or La (α-La mAb) surface staining in non-permeabilized, human osteoclasts 3 days after RANKL addition before or after EGTA incubation. **(h)** Quantification of surface fluorescence intensity from **g**. (Anx A5 and RANK n = 2; La n = 3) (P = 0.02, 0.04 and 0.3, respectively) **(i)** Quantification of surface fluorescence intensity of Anx A5, La or RANK treated or not treated with 60 μM A01 (n = 3) (P = 0.006, 0.01 and 0.35, respectively). In **e, h** and **I**, ~100 cells were assessed per each condition in each independent experiment. **(d, e, f, h, i)**. Data are presented as mean values +/- SEM. Statistical significance was assessed via one-tailed paired t-test. * = P < 0.05; ** = P < 0.001. Source data are provided as a Source Data file.

osteoclasts at the membrane fusion stage rather than some pre-fusion stage of differentiation. To this point, application of recombinant La to LPC-synchronized osteoclasts promotes osteoclast membrane fusion (Fig. 4e).

All these data indicated that La functions at the cell surface during the membrane fusion stage of osteoclast formation.

To test whether La at the cell surface interacts with some other protein(s) involved in fusion, we assessed whether La interacted with Anx A5, a peripheral membrane protein, also involved in membrane fusion stage of osteoclast formation and upregulated at similar time-points in osteoclastogenesis[18]. We immunoprecipitated La and La-containing protein complexes from fusing human osteoclasts on magnetic beads with α-La mAbs and found that La protein complexes

contained Anx A5 (Fig. 5a). La supramolecular complexes from fusing osteoclasts contained neither Anx A1 nor Anx A4, both abundant in fusing osteoclasts (Fig. S6a), demonstrating specificity in La's association with Anx A5.

In further support of direct La-Anx A5 binding, we found that streptavidin pull-down of Biotin-Anx A5 from the mixture of recombinant 6xhis-La and Biotin-Anx A5, but not unrelated protein Biotin-Actin, also enriched 6xhis-La in contrast to 6xhis-La alone, suggesting that La and Anx A5 bind one another in the absence of other proteins (Fig. 5b, Fig. S6b). Interestingly, pull-down experiments suggest that both La1-187 and La188-375 have some affinity for Anx A5 (Fig. S6c, d), suggesting that the La-Anx A5 binding site includes residues in both regions, or, alternatively, that La has more than one Anx A5 binding site.

Anx A5 binds to cell membranes and phospholipid bilayers in a $Ca^{2+}$ and phosphatidylserine (PS) dependent manner[48]. We hypothesized that Anx A5- La interactions can explain how soluble protein La binds to the membranes. We tested this hypothesis by adding recombinant La alone or along with recombinant Anx A5 to multilamellar liposomes (Fig. 5c, d and Fig. S6e–j) and then pelleted the liposomes with bound proteins by centrifugation. To measure equilibrium binding, we split the initial volume into top fraction, containing mostly proteins not bound to liposomes, and bottom fractions, containing unbound proteins and proteins bound to the pelleted liposomes. The pellet was then resuspended in the total volume of the bottom fraction and samples from each fraction were used to measure protein. Qualitatively for proteins that do not bind to liposomes, we expect equal distribution between top and bottom fractions, while for proteins binding to liposomes, we expect enrichment in the bottom fraction. The fraction of the protein bound to the liposomes can be calculated as described in the Materials and Methods. While La alone did not pellet along with PS-containing liposomes in the absence of $Ca^{2+}$, both La and Anx A5 were enriched with liposomes in response to $Ca^{2+}$ and La's association with liposomes was greatly increased with addtion of Anx A5 (Fig. 5d, S6e). La membrane association depended on Anx A5, $Ca^{2+}$ and PS, as neither La nor Anx A5 were enriched with liposomes lacking PS (Fig. 5d, S6e). In control experiments, we verified that neither La nor Anx A5 notably changed the efficiency of liposome pelleting and that La association with PS-containing liposomes was not observed when Anx A5 was replaced with the unrelated protein Actin (Fig. S6f–j). Finding that the pelleting of La is greatly enriched by its association with Anx A5 in a PS- and $Ca^{2+}$-dependent manner suggests that La-Anx A5-PS interactions can anchor La to osteoclast membranes.

The hypothesis that La binding to the surface of differentiating osteoclasts involves Anx A5 has been further supported using three complementary experimental approaches. In the first approach, we found that RNAi suppression of Anx A5 expression lowered cell surface La and inhibited osteoclast fusion (Fig. 5e, f). These findings confirmed the importance of Anx A5 for osteoclast fusion (see also[18]) and suggested that the enrichment of La at cell surface involves Anx A5. In the second approach, we took advantage of the dependence of Anx A5-PS binding on $Ca^{2+}$. We found that 10-min incubation of fusing human osteoclasts with complete medium supplemented with the $Ca^{2+}$ chelator EGTA, removes endogenous Anx A5 bound to the surface of fusing osteoclasts (Fig. 5g, h). This loss of Anx A5 is accompanied by a comparable loss of surface-bound La but has no effect on the transmembrane receptor RANK. In the third approach, we lowered the amounts of Anx A5 at the surface of human osteoclasts by suppressing the externalization of PS in fusing osteoclasts. PS is externalized on the surface of fusing osteoclasts by TMEM16 lipid scramblases[18]. Inhibition of TMEM16-dependent PS exposure with TMEM16 scramblase inhibitor A01[18,49] dramatically lowered the amounts of Anx A5 on the surface of fusing osteoclasts and resulted in a comparable decrease of surface La (Fig. 5i). Again, this treatment had no effect on the surface RANK receptor. Moreover, both calcium depletion and A01 application have been reported to inhibit osteoclast fusion[18,50]. Our findings suggest that direct interactions between La and extracellular Anx A5 enrich La at sites of transient PS exposure on the surface of osteoclasts at the time of fusion[18], facilitating La association with the surface of fusing osteoclasts.

Proteins involved in membrane fusion can be divided into protein fusogens that are sufficient for generating hemifusion intermediates and opening of fusion pores, and proteins that regulate fusogen activity[15]. To test whether cell surface La may fuse membranes on its own, functioning as an active protein fusogen, we assessed La's ability to promote fusion between 3T3 fibroblasts, stably expressing HA0 (an uncleaved form of the influenza fusogen hemagglutinin (HA)), and red blood cells (RBCs) labeled with lipid and a content probes[51]. HA0 is fusion-incompetent but establishes very tight contacts between HA0-

expressing fibroblasts and RBCs. As seen in Figure S7a (top panel), La application did not induce fibroblast-RBC fusion. None of 872 analyzed HA0-cell bound RBCs exchanged lipid (hemifusion indicator) or cytoplasmic (fusion pore indicator) probes in response to application of 40 nM recombinant La. Based on Wilson's method[52], the probability of La-mediated fibroblast-RBC fusion was estimated to be lower than 0.0044 per cell contact. In contrast, when we cleaved HA0, via trypsin, into a fusion-competent HA, low pH application triggered robust fusion with the probability of fusion exceeds 0.5 per fibroblast-RBC contact (Fig. S7a, bottom panel and[51]). Thus, if La has any fusogenic activity on its own, this activity is at least 100-fold lower than that of a bona fide fusogen - activated HA.

In another experimental approach, we tested whether application of recombinant La induces lipid mixing between liposomes labeled with a FRET pair of fluorescent lipids and unlabeled liposomes (Fig. S7b). In this approach, lipid mixing dilutes the probes and increases the fluorescence of the donor probe. Addition of Anx A5, La and 2 mM $Ca^{2+}$ to PS-containing liposomes (conditions found in Fig. 5d to support La binding to the liposomes) had no effect on fluorescence dequenching, arguing against fusogenic activity of PS-bound La-Anx A5 complexes. In contrast, addition of 5 mM $Ca^{2+}$ alone, as expected[53], induced efficient lipid mixing between PS-containing liposomes (positive control).

PS exposure and Anx A5 are known to promote fusion in other cell-cell fusion systems[15]. To test whether the dependence on cell surface La is specific to osteoclasts, we examined whether myoblast fusion is associated with changes in the expression and localization of La, and relies on this protein.

As expected for muscle cells[54], we found that C2C12 cells express La. La was expressed at similar levels in proliferating and differentiating cells (Fig. S7c). We immunostained permeabilized differentiating myoblasts for La and found it to be distributed throughout the cells, in contrast to being concentrated at the rim of the cells, as in differentiating Raw 264.7 cells (Fig. S7d). In non-permeabilized cells, we found some La at the surface of C2C12 cells, but both the surface staining and the ratio of fluorescence intensities for surface La (non-permeabilized cells) vs cytosolic La (permeabilized cells) was considerably lower for C2C12 cells than for Raw 264.7 cells (Fig. S7d, e). We then examined whether myoblast fusion, like osteoclast fusion, can be inhibited by reagents targeting the activity of the cell surface La. Neither α-La mAb nor recombinant La significantly affected the fusion of C2C12 myoblasts (Fig. S7f, g). C2C12 cells also demonstrated a considerably lower ability to bind exogenously applied recombinant La protein (Fig. S7h). Finding that myoblast fusion is neither accompanied by the same changes in La expression and localization as osteoclast fusion, nor influenced by the same surface La-targeting reagents strongly suggests that surface La's role in osteoclasts is not a conserved mechanistic motif shared by all cell-cell fusion processes.

To summarize, while La on its own does not initiate fusion in heterologous systems, cell surface-associated La promotes membrane fusion in the formation of multinucleated osteoclasts. This role of La is independent of several domains previously characterized for their functions in La's canonical role in RNA metabolism, including its La domain, RRM1 and NLS. The association of La with the cell surface of osteoclasts depends on PS externalization and direct interactions between La, Anx A5 and PS.

## La presents a potential target for influencing osteoclast formation and function

The previously elucidated relationship between osteoclast fusion and bone resorption[8–10] led us to hypothesize that by regulating osteoclast size, La also regulates bone resorption. We tested this hypothesis by differentiating osteoclasts on fluoresceinated calcium phosphate, a biomimetic of bone, and assessed osteoclast-dependent bone resorption by the release of fluorescein into the media (illustrated in

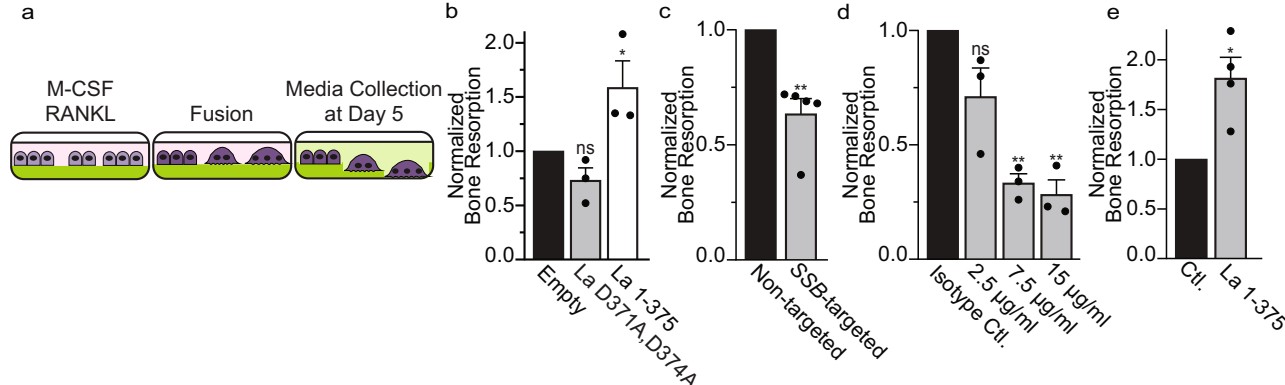

**Fig. 6 | Interfering with La influences bone resorption by human osteoclasts.** **a** Illustration depicting the use of fluoresceinamine-labeled chondroitin sulfate trapped in calcium phosphate coated plates to assay bone resorption. **b** Bone resorption in osteoclasts overexpressing uncleavable La (D371A, D374A) or truncated La (1-375) normalized to that for the osteoclasts transfected with empty plasmid. ($n = 3$) ($P = 0.06$ and 0.05). **c** Bone resorption in osteoclasts transfected with siRNA targeting La transcript normalized to that for non-targeted siRNA.

($n = 5$) ($P = 0.005$). **d** Bone resorption in osteoclasts exposed to different concentrations of α-La mAb normalized to that for the cells treated with isotype control IgG at 7.5 μg/ml. ($n = 3$) ($P = 0.0723$, 0.0098 and 0.004). **e** Bone resorption in osteoclasts treated with 40 nM recombinant La 1-375 normalized to the control after adding the same amount of PBS. ($n = 4$) ($P = 0.03$) **b**–**e** Statistical significance was assessed via one-tailed paired t-test. Data are presented as mean values +/- SEM. Source data are provided as a Source Data file.

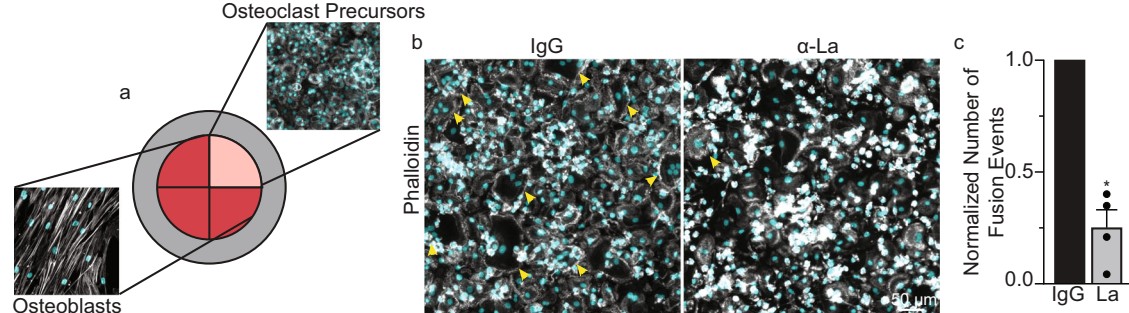

**Fig. 7 | Osteoclast formation in human osteoblast/osteoclast precursor co-culture depends on La protein.** **a** Schematic of the multi-well configuration of the human osteoblast/osteoclast co-culture system used. Multi-well dividers were removed following the M-CSF derivation of human monocytes, and osteoclast precursors and primary osteoblast media intermixed. (Grey=Phalloidin-Alexa488, Cyan=Hoechst). **b** Representative immunofluorescence images comparing

osteoclast fusion in the presence of 6 μg/ml control IgG or α-La mAb antibodies. Arrowheads denote syncytia with ≥3 nuclei. **c** Quantification of osteoclast fusion in the co-cultures in the presence of α-La mAb normalized to that in the presence of control IgG. ($n = 4$) ($P = 0.03$) Statistical significance was assessed via one-tailed paired t-test. Data are presented as mean values +/- SEM. Source data are provided as a Source Data file.

Fig. 6a). Monocyte-derived precursors (only M-CSF) released minimal trapped fluorescein, but the addition of RANKL resulted in formation of multinucleated osteoclasts that readily resorbed calcium phosphate and released fluorescein (Fig. S1b). Overexpression of La 1-375 promoted bone resorption, while the uncleavable La mutant, D371,374 A, had no effect (Fig. 6b). Moreover, RNAi-mediated reduction of La reduced bone resorption by ~40% compared to non-targeted controls (Fig. 6c). The α-La mAb that inhibits fusion (Fig. 3f) also dramatically reduced osteoclast-dependent bone resorption in a dose-dependent manner (Fig. 6d). Finally, the extracellular addition of recombinant La 1-375 to fusing osteoclasts dramatically increased osteoclast bone resorption (Fig. 6e). From these data, we conclude that targeting cell surface La bidirectionally regulates both osteoclast fusion and subsequent bone resorption.

In biologically relevant situations, osteoclastogenesis develops in the context of interactions between osteoclast precursors with bone-forming osteoblasts/osteocytes and other cell types, generating RANKL and many other osteoclastogenesis-regulating factors[55]. To explore whether La is involved in osteoblast-induced osteoclast formation, we co-cultured primary human osteoblasts isolated from trabecular bone and human osteoclast precursors, derived via M-CSF

induction of primary human monocytes. Osteoblasts and osteoclast precursors were cultured isolated from each other by well inserts (Fig. 7a). Without removing well inserts, we observed no fusion between osteoclast precursors. Upon removal of well inserts, media from the osteoblast/osteoclast wells mixed and co-cultured osteoclast precursors rapidly fused to produce multinucleated osteoclasts. Addition of α-La mAb antibody blocked nearly 75% of the fusion between osteoclasts in such co-cultures (Fig. 7b, c) confirming the involvement of La in osteoclast formation in a biologically relevant model of bone remodeling lesions.

To explore whether La function plays a role in bone pathology, we have focused on fibrous dysplasia of bone (FD), an osteoclast-dependent bone disease[56]. FD is caused by gain-of-function mutations in Gαs that lead to constitutively increased cAMP signaling and upregulation of cAMP/RANKL-dependent osteoclastogenesis[57]. In a conditional, tetracycline-inducible mouse model, FD-like bone lesions, develop in adult mice within 2 weeks following doxycycline (Doxy) administration[58]. The formation of these lesions is driven by activation of an inducible gain-of-function mutant, Gαs^R201C, specifically in cells of the skeletal stem cell linage responsible for the excessive RANKL production observed in FD. This excessive RANKL production results

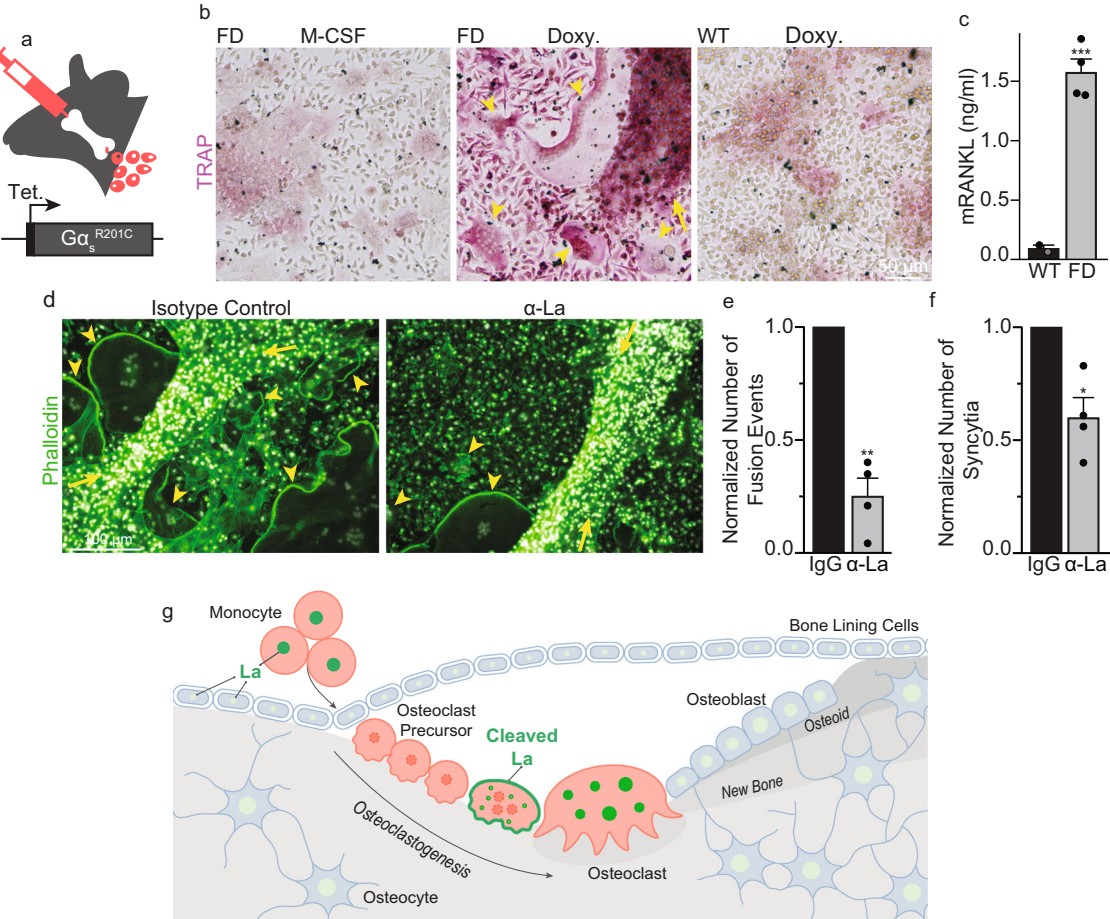

**Fig. 8 | α-La treatment suppresses ectopic osteoclast formation in fibrous dysplasia explants. a** Illustration of ex vivo bone marrow culture system based on a tetracycline-inducible model of FD. **b** TRAP staining of bone marrow explants from a homozygous $Ga_s^{R201C}$ mouse (FD) and a wild-type littermate (WT). Explants were either cultured with M-CSF alone, or M-CSF and Doxy. **c** ELISA quantification of mRANKL produced in FD vs WT cultures following Doxy treatment. (WT $n = 2$; FD $n = 4$) ($P = 0.0005$). **d** Representative images of FD explants activated by Doxy treated with 6 μg/ml isotype control or α-La mAb. **e** Quantification of the number of fusions producing osteoclast syncytia with ≥3 nuclei. ($n = 4$) ($P = 0.005$) **f** Quantification of the number of osteoclast syncytia with ≥3 nuclei from **d**. ($n = 4$) ($P = 0.02$) In **b** and **d**, arrowheads denote multinucleated osteoclasts and arrows denote fibrous cell masses developed after Doxy addition and characteristic for FD. Statistical significance was assessed via one-tailed unpaired t-test in **c** or paired,

t-tests in **e** and **f**. Data are presented as mean values + /- SEM. **g** An illustration of changes in the steady-state level and cellular localization of La protein in the process of osteoclast formation. La (green) carries out its canonical, ancient function in the nuclei of all eukaryotic cells as an essential RNA-binding protein. We propose that La has an additional, specialized function in the formation of multinucleated osteoclasts. First, La dissipates as circulating monocytes become osteoclast precursor cells. When osteoclast commitment is initiated by RANKL, La returns but is quickly cleaved by proteases and shuttled to the surface of osteoclasts. At the surface of fusing osteoclasts, La plays a novel role as a membrane fusion manager. When osteoclasts arrive at the "right size" for their biological function, surface La dissipates and is replaced by canonical, non-cleaved La that returns to the nuclei of mature osteoclasts. Source data are provided as a Source Data file.

in the ectopic formation of numerous, large osteoclasts that excessively erode healthy bone. Using bone marrow explants from these FD mice, we established a robust, ex vivo model of the ectopic osteoclast formation observed in FD (described in the Methods and illustrated in Fig. 8a). As depicted in Fig. 8b, the culture of these FD explants in the presence of M-CSF alone resulted in numerous adherent cells but no multinucleated, tartrate-resistant acid phosphatase positive (TRAP)⁺ osteoclasts. In contrast, addition of Doxy resulted in the rapid development of fibrous cell clumps (arrow) and numerous multinucleated, TRAP⁺ osteoclasts (arrowheads) that were not observed in explants from wild-type littermates, lacking the inducible $Ga_s^{R201C}$ element. Doxy-induced osteoclastogenesis was accompanied by a ~17-fold increase in mRANKL produced by the explants (Fig. 8c). Importantly, α-La mAb blocked osteoclast fusion elicited by the addition of Doxy to FD explants by ~60% and reduced the number of multinucleated osteoclasts observed by ~40% (Fig. 8d–f).

To summarize, cell-surface La regulates the formation of human and murine multinucleated osteoclasts triggered by biologically

relevant interactions between osteoclast precursors and bone-forming cells. Targeting La modulates fusion between osteoclast precursors and, in turn, alters the propensity of the resulting osteoclasts to resorb bone. Furthermore, our findings that La is involved in osteoclast formation in an ex vivo FD model suggest that targeting La function at the surface of developing osteoclasts can be an effective therapeutic intervention in FD, and likely other resorptive bone diseases stemming from excessive osteoclast activity.

## Discussion
Here we report that the differentiation of murine and human monocytes into multinucleated osteoclasts is dependent on tightly choreographed changes in the steady-state level, post-translational modification, and cellular localization of La (Fig. 8g). In fusing osteoclasts, La is found at the surface of the cells, and this cell-surface associated La, rather than intracellular La, regulates osteoclast fusion. The changes in the expression of La and the functional role of its cell surface form in osteoclastogenesis are unexpected in the context of

the vast literature covering La's role in RNA biology. La is generally thought of as an abundant, ubiquitous, mostly phosphorylated RNA-binding protein largely confined to the nucleus in virtually all eukaryotic cell types[26,41]. However, at the onset of osteoclastogenesis, M-CSF-derived precursors show a dramatic loss of La protein, suggesting that this differentiation process may require the concerted down-regulation of a specific La-regulated pool of mRNAs triggered by the loss of steady state La. The negligible levels of La protein in osteoclast precursors despite abundant transcript could be the result of translational inhibition, the sequestration of these transcripts (e.g., into RNA speckles) or the rapid degradation of La protein within osteoclast precursors. In the following RANKL-induced stages of osteoclastogenesis, La reappears as a non-phosphorylated, proteolytically cleaved species in the cytoplasm and at the surface of the fusing osteoclast precursors. When the growth of osteoclasts slows, in the late stages of fusion, La is observed at its conventional molecular weight and nuclear localization. The rate of formation, the sizes of multinucleated syncytia and the subsequent bone resorption activity of osteoclasts are regulated by cell-surface La protein. In fact, cell-surface La regulates osteoclast functions by modulating the membrane fusion stage of osteoclast formation, not upstream differentiation processes. Lowering the amount of La by suppressing the steady-state level of its transcript, blocking its proteolytic processing, or inhibiting its activity with antibodies inhibits fusion. Conversely, increasing La's steady-state concentration by either overexpression or application of recombinant protein promotes fusion. In addition, the addition of α-La antibodies or recombinant La at the surface of osteoclasts inhibits and promotes synchronized osteoclast fusion, respectfully. These effects occur in 90 minutes and minimize the possibility that our antibodies/recombinant proteins are taken up by osteoclasts, escape their endocytic compartments and exert their effects in the cytoplasm. Importantly, the upregulation of cell surface La and its involvement in osteoclast fusion have been observed for both primary human osteoclasts and for murine Raw 264.7 cells. Despite subtle differences in the levels and localization of La noted above, which may be due to differences between primary and immortalized cells, or to using antibodies raised exclusively to human La, our data strongly suggest that La's role in regulating fusion is conserved in mammals. In summary, our work demonstrates that La, a key protein in the RNA biology of eukaryotic cells, lives a second life at the surface of osteoclasts where it moonlights as a master regulator of osteoclast membrane fusion.

Our data demonstrate that La's role in regulating osteoclast fusion and bone resorption is separate from the well-described canonical functions of La in RNA metabolism and represents a novel function for La protein. First, our ability to quickly inhibit or promote synchronized osteoclast membrane fusion by non-membrane permeable reagents (e.g., antibodies, recombinant La) indicates that the regulation of osteoclast fusion depends on La at the surface of osteoclasts. In contrast, the well-characterized functions of La in the processing and metabolism of a variety of different RNAs[26,27] and in sorting microRNAs into extracellular vesicles[59] are carried out in the nucleus or cytoplasm and depend on La domain-RNA interactions. Only in some special biological processes, including in herpes simplex virus and adenovirus infections[60,61], adding serum to serum-starved cells[32,62] and in the early stages of apoptosis[33,42], is La protein found exposed at the surface of the viable cells[33,42]. However, the only suggested function of these previous examples of cell surface-bound La has been to recruit regulatory T cells to damaged tissues to downregulate an immune response during cell death[63].

In another striking distinction from the well-characterized functions of La in RNA metabolism, La regulation of osteoclast formation does not depend on interactions between the highly conserved La domain or RRM1 with RNA. This conclusion is supported by our finding that neither mutations in critical residues within the La domain nor deleting the entire N-terminal half of La protein (containing both the La

domain and RRM1) abolishes the ability of recombinant La to promote osteoclast fusion. Since recent studies found RNAs present on the surface of living cells that are involved in monocyte interactions[64,65], it remains possible that RNA is involved in surface La's role in regulating osteoclast fusion. However, even if the function of La in osteoclast formation depends on La-RNA interactions at the cell surface through some yet unknown mechanism, possibly involving RRM2[30], this function fundamentally differs from the classical functions of La dependent on its La domain- and RRM1- mediated RNA binding in the nucleus and cytoplasm.

The mechanisms by which cell-surface La regulates osteoclast fusion remain to be clarified. Since La, on its own, initiates neither hemifusion nor fusion between bound membranes, it is unlikely that La directly catalyzes and/or drives membrane fusion. More likely La recruits or stimulates other components of the osteoclast fusion complex. The latter scenario is supported by our findings that highlight La's association with the fusion regulator Anx A5. Anx A5 has been implicated in several cell-cell fusion processes (reviewed in[66]). In the case of osteoclast fusion, osteoclastogenic differentiation of human monocytes is associated with a strong increase in the amount of Anx A5 present at the cell surface and treatments suppressing the expression and activity of cell surface Anx A5 inhibit synchronized osteoclast fusion[18]. In contrast to Anx A5, two other members of the Annexin protein family: Anx A1, which is not involved in human osteoclast fusion[18], and Anx A4, which has many structural and functional similarities to AnxA5[67], were not found in supramolecular complexes with La. The lack of Anxs A1 and A4 but the presence of Anx A5 in La protein complexes further supports the hypothesis that La specifically associates with partners involved in the fusion stage of osteoclast formation. Specific mechanisms by which interactions between La and Anx A5, and possibly, La interactions with other components of osteoclast fusion machinery, regulate cell-cell fusion remain to be clarified. We find that recombinant La and Anx A5 directly interact, and that Anx A5 enriches La on membranes containing PS in a $Ca^{2+}$-dependent manner. These observations explain how La, a soluble protein, associates with cell membranes, connect La function in osteoclastogenesis with the non-apoptotic PS exposure signaling pathway suggested to trigger osteoclast fusion[66] and, in combination with the previously reported dependence of osteoclast fusion on cell surface PS and Anx A5[18], shed additional light on how osteoclasts employ PS to initiate the assembly of a fusion complex between committed precursors. Note that the established link between fusion efficiency and cell surface concentrations of La and PS-bound Anx A5 at the surface of fusion-committed osteoclasts remains correlative and only future work uncovering molecular mechanisms of osteoclast fusion will clarify the specific role of these proteins in the fusion pathway.

Our finding that La regulates osteoclast fusion but not myoblast fusion, also dependent on PS exposure[68,69] and cell surface Anx A5[70], suggests that La is an osteoclast-specific fusion regulator and not some general factor impacting all cell-cell fusions. Low efficiency of the binding of exogenously added La to Anx A5 at the surface of myoblasts may indicate that these Anx A5 molecules are bound to another, yet-unidentified regulator of myoblast fusion. Future work will clarify whether myoblast fusion and other cell-cell fusion processes depend on proteins that, like La in osteoclasts, bind PS-bound Anxs at the surface of fusion-committed cells connecting PS signaling regulation with cell type-specific components of the fusion machines[66].

While we found La to promote and regulate the formation of multinucleated osteoclasts, we do not know yet whether La is indispensable for osteoclast formation. La is required for mouse development and the establishment of embryonic stem cells[38], and efforts to specifically delete La in B cell progenitors and the forebrain have resulted in the loss of these cell types, suggesting that targeted knockout approaches are unlikely to resolve whether La is required for

osteoclast formation[37]. Importantly, while the steady-state levels of La are quite low in osteoclasts precursors, these levels can be sufficient to support RNA metabolism. Further resolution of the regions within La that are critical for its role in osteoclast fusion may afford us the ability to perturb La's fusion function, while leaving its essential, RNA chaperone functions intact. This resolution will be essential for testing whether La is essential for osteoclast membrane fusion in future studies.

Our findings add La to a growing list of "moonlighting proteins" that serve several, sometimes strikingly unrelated, functions[71,72]. For instance, the ubiquitous housekeeping proteins glycolytic enzyme Glucose-6-phosphate isomerase and RNA-binding protein nucleolin that localized primarily in the cytoplasm and nucleolus, respectively, have second, unrelated functions at the cell surface[71,73]. For La specifically, the N-terminal La motif is highly conserved from yeast to man[26]. However, latter domains of La, particularly the C- terminal half of the protein, are only weakly conserved. To this point, from yeast to vertebrata La experienced ~50% increase in its molecular mass due to an expansion in its C-terminus. Our data suggest that while the La motif, conserved throughout a billion years of evolution, is responsible for La's original functions in RNA metabolism, regions in La's C-terminal expansion, yet unidentified, carry out the protein's more recently acquired function in regulating osteoclast formation.

The specific contributions of the different regions of La protein to its different functions; the evolutionary processes by which La protein has acquired its role in the formation of multinucleated osteoclasts; and the mechanisms that control the delivery of NLS-lacking La species to the surface of cells in a temporally appropriate manner are exciting questions we hope to address in future work. Intriguingly, La is not the only RNA-binding protein implicated in cell-cell fusion. Both Acheron, a member of La protein family[74], and HuR, a member of ELAV-family of RNA-binding proteins[75], are involved in the formation of multinucleated myotubes. Comparison of the contributions of different RNA chaperones to diverse cell fusion processes will clarify whether cell fusion control by moonlighting RNA-binding proteins represents a conserved functional motif that cells use to create syncytial cell types.

Imbalance of bone-formation and resorption in many skeletal diseases is linked to either excessive (e.g., osteoporosis, Paget's disease and FD), or insufficient activity of osteoclasts (e.g., osteopetrosis). Here we report that the formation of multinucleated, human osteoclasts can be inhibited or promoted by treatments targeting La protein at the surface of osteoclast precursors. Importantly, α-La mAb inhibited fusion and bone resorption by osteoclasts derived from RANKL-activated monocytes. α-La mAb treatment also inhibited the formation of multinucleated osteoclasts in human osteoclast precursor/osteoblast co-cultures, where osteoclastogenic factors are produced by osteoblasts within the lesion[76]. Moreover, the hypothesis that cell-surface La plays an important role in osteoclast formation within biologically relevant contexts is further substantiated by our experiments in an ex vivo model of FD. Development of FD is characterized by drastically increased levels of RANKL and other osteoclastogenic factors in serum and the excessive, ectopic formation of numerous multinucleated osteoclasts in the vicinity of bone lesions. As expected, induction of the FD phenotype in bone marrow explants resulted in high concentrations of RANKL and ectopic osteoclastogenesis. Finding that α-La antibody inhibits the formation of multinucleated osteoclasts, both in size and number, suggests the involvement of La in bone pathologies and highlights the potential of La as a target for future therapeutic development.

Some of the proteins involved in the early stages of osteoclastogenic differentiation have already been tested in animal and/or clinical studies as potential therapeutic targets[77]. The α-RANKL antibody denosumab is an FDA-approved drug for the treatment of osteoporosis[21,78]. La-dependent osteoclast fusion, downstream of the RANKL/RANK/osteoprotegerin signaling pathway, presents a potential target for therapies at a different mechanistic stage of bone remodeling. Considering that mononucleated osteoclasts do resorb bones, blocking La-dependent osteoclast fusion is expected to have more subtle and selective effects on bone resorption than blocking the upstream formation of osteoclast precursors all together with α-RANKL antibodies[79–81]. Like RANKL, cell surface La is accessible for cell-impermeable drugs. In some clinical situations, the more subtle action of La-targeting treatments can be advantageous. Furthermore, unlike RANKL, which in addition to osteoclastogenesis regulates immune response[82], the only known function of cell surface La is its newly identified role in regulating osteoclast fusion. Surface La's specificity may minimize off-target effects. Finally, osteoclasts are known to release factors that regulate osteoblast activity[83]. Blocking osteoclastogenesis altogether by targeting RANKL likely blocks osteoclast-osteoblast signaling. Suppressing the fusion stage of osteoclast formation by targeting La, while maintaining the ability of osteoclasts to differentiate, may maintain this osteoclast-osteoblast crosstalk within the bone remodeling lesion.

In summary, our data demonstrate a function of La protein as a key regulator of osteoclast formation, a role strikingly different in place of action, mechanism, and partner protein (Anx A5) from its well-recognized functions as an RNA-chaperone. The future development of safe and effective reagents targeting cell-surface bound La may lead to novel antiresorptive therapies for osteoporosis, mechanistically orthogonal to the existing approaches. We expect our new ex vivo resorptive bone disease model utilizing FD cells, which condensed to days the time course of excessive osteoclast formation typical of disease, to represent a powerful tool for the study of osteoclastogenesis in health and disease states.

## Methods

### Reagents

Human M-CSF and RANKL and murine M-CSF and RANKL were purchased from Cell Sciences (catalogue #CRM146B; #CRR100B; # CRM735B and CRR101D, respectively). LPC (1-lauroyl-2- hydroxy-sn-glycero-3-phosphocholine, #855475); PC (1,2-dioleoyl-sn-glycero-3-phosphocholine, #850375 C); PS (1,2-dioleoyl-sn-glycero-3-phospho-L-serine, #840035 C); lissamine rhodamine phosphatidylethanolamine (1,2-dioleoyl-sn-glycero-3-phosphoethanolamine-N-(lissamine rhodamine B sulfonyl, #810150 C) were purchased from Avanti Polar Lipids. Bone Resorption Assay Kits were purchased from Cosmo Bio Co. (Catalogue # CSR-BRA-24KIT) and used according to the manufacturer's instructions. Hoechst 33342 and phalloidin-Alexa555 were purchased from Invitrogen (#H3570 and A30106, respectively). TRAP staining reagents were purchased from Cosmo Bio Co. (#PMC-AK04F-COS). The fluorescent lipid PKH26 (PKH26GL-1KT) and carboxyfluorescein, CF (5-(and-6)-Carboxyfluorescein, mixed isomers, #C368) were purchased from Sigma and Invitrogen, respectively. Calcium-activated chloride channel inhibitor CaCCinh-A01 ("A01") used to suppress TMEM16 scramblase[18,49], the pan-caspase inhibitor z-vad-fmk and the caspase 3 inhibitor z-DEVD-fmk were purchased from Tocris (#4877, #2163 and #2166, respectively). Ethylene glycol-bis(2-aminoethylether)-N,N,N',N'-tetraacetic acid (EGTA) was purchased from Sigma-Aldrich (#03777). Molecular weight standards used were purchased from BioRad (1610374) and Biodynamics (DM660). Membrane-permeant Green CMFDA cell tracker and orange CMRA Cell Tracker were purchased from ThermoFisher Scientific (#C7025 and #C34551, respectively). CF dye protein labeling kits were purchased from Biotium (#92213 and #92214).

### Animals

All animal studies were carried out according to NIH-Intramural Animal Care and Use Committee (ACUC) of the National Institute of Dental and Craniofacial Research approved protocols (ASP #19-897), in compliance with the Guide for the Care and Use of Laboratory Animals.

A mouse model of fibrous dysplasia with inducible expression of hyperactive $G\alpha_s^{R201C}$ in cells of the osteogenic lineage[56] was used to obtain bone marrow explants (described below). For this study we used 12–18-week-old females generated by genetic cross of C57BL/6 and FVB/N strains. Mice were separated by sex at weaning and housed in shared cages of maximum 5 littermates in a conventional veterinary facility (with quarantine requirements, and exclusion of specific pathogens) with a 12 h/12 h light-dark cycle and fed ad libitum with NIH 07 (autoclavable) hard diet (Envigo, Frederick, MD). Mice colonies health was monitored using standard observation methods and sentinel cages that were periodically tested for pathogens. Mice were euthanized by $CO_2$ inhalation, and cervical dislocation was performed to ensure death after no external signs of breathing were noticeable.

### Murine bone marrow explant culture

The tibia and femur were dissected from an inducible murine model of fibrous dysplasia described in[56] or wild-type littermates. Holes were drilled into the epyphyses of each bone using a 22-gauge hypodermic needle, and the bone marrow was flushed into a culture dish using alpha MEM. These bone marrow isolates were further dissociated through a fresh 22-gauge hypodermic needle to obtain a single cell suspension, and cultured in alpha MEM plus 20% FBS, 1x pen/strep and 1x Normocin (InvivoGen, # Ant-nr-1) for 7 days in T-75 culture flasks. Cells that adhered to the flask were washed 3 times with PBS and passaged using 0.05% Trypsin and a cell scraper and cultured for up to 3 passages in alpha MEM plus 20% FBS and 1x pen/strep. For $G\alpha_s^{R201C}$ expression induction in the bone marrow stromal cell subset of the explants, cells were plated at near confluency in 6-well plates and treated with 1 μM doxycycline (Sigma, # D9891-5G). During induction, media were refreshed daily. For antibody treatment, antibodies were added overnight when initial cell-cell fusion was observed (typically between 2–3 days of doxycycline treatment) at 6 μg/ml overnight.

### Cell cultures

**Osteoclasts.** Elutriated monocytes from healthy donors were obtained through the Department of Transfusion Medicine at National Institutes of Health under protocol 99-CC-0168 approved by the National Institutes of Health Institutional Review Board. Research blood donors provided written informed consent and blood samples were de-identified prior to distribution, Clinical Trials Number: NCT00001846. We also used elutriated monocytes from healthy donors obtained through Elutriation Core Facility, University of Nebraska Medical Center, informed consent was obtained under an Institutional Review Board approved protocol for human subject research 0417-22-FB. Research blood donors provided informed consent and samples were de-identified prior to distribution. Elutriated monocytes were plated at ~ $2.9 \times 10^5$ per $cm^2$ in 35 mm dishes with polymer coverslip bottoms (Ibidi #81156) for imaging or 35 mm or 10 cm dishes for biochemical experiments in complete media [α-MEM supplemented with 10% Fetal Bovine Serum (FBS) and penicillin-streptomycin-L-glutamine (Gibco Invitrogen # 12571063; #26140079 and #10378016, respectively)]. Monocytes were differentiated to M2 macrophages in the presence of 100 ng/ml M-CSF for 6 days and then differentiated with 100 ng/ml M-CSF and 100 ng/ml RANKL for 3 days unless indicated otherwise. RAW 264.7 cells (ATCC, Manassas, VA, # TIB-71) were maintained in DMEM supplemented with 10% FBS to a maximum of 8 passages. RAW 264.7 cells were differentiated to osteoclasts in the presence of 100 ng/ml murine RANKL for 5 days. To separate unfused mononucleated and fused, multinucleated RAW 264.7 cells into separate fractions we have taken advantage of the much stronger adherence of multinucleated cells to culture dish plastic. After washing in PBS, mixed RAW 264.7 cultures (following RANKL differentiation) were left in $Ca^{2+}$ and $Mg^{2+}$ free PBS for 10 min at room temperature. Culture dishes were then tapped on the lab bench and a large portion of unfused, mononucleated RAW264.7 cells were

released, and these released cells were collected via centrifugation. This process was repeated 2–4 times until dishes were left with a population of primarily fused, multinucleated syncytia. Mononucleated and multinucleated cell fractions were then processed for biochemical, or imaging experiments as described below.

**Human Osteoblast/Osteoclast Co-culture.** Osteoblasts isolated from the trabecular bone of healthy individuals with the informed consent of the donors were obtained from PromoCell (#C12720), operating under Approval # 219-04 of the Ethics Commission of the State Chamber of Medicine. The cells were cultured according to the manufacturer's instructions. Osteoclast precursors were derived from primary human monocytes by 6 days of culture in M-CSF as described above. Osteoblasts and osteoclast precursors were cultured in 35 mm dishes with 4-well culture inserts (ibidi, #81156) at a 3:1 well ratio. 48 h before co-culture mixing, osteoblasts were switched to serum free alpha MEM with 1x pen/strep (Gibco). Following serum starvation, 4-well culture inserts were removed, and cells were cultured in their conditioned media overnight with or without treatment. Cells were fixed with 4% paraformaldehyde the following morning.

**HAO-expressing cells and RBCs.** NIH 3T3 mouse fibroblasts of clone 15 cell line that stably express influenza were a kind gift of Dr. Joshua Zimmerberg, NICHD, NIH[84]. These HAO-expressing cells were cultured at 37 °C and 5% $CO_2$ in DMEM supplemented with 10% heat-inactivated FBS and antibiotics. The cells were used without trypsin pretreatment to keep HA in a fusion-incompetent form. In positive control experiments, HA-cells were treated with 5 μg/ml trypsin for 5 min at room temperature. Human red blood cells (RBCs) from healthy donors were obtained through the Department of Transfusion Medicine under protocol 99-CC-0168 approved by the National Institutes of Health Institutional Review Board. Research blood donors provided written informed consent and blood samples were de-identified prior to distribution, Clinical Trials Number: NCT00001846. RBC were labeled with the fluorescent membrane dye PKH26 and loaded with the fluorescent water-soluble dye CF. HA-cells were washed twice with PBS and then incubated for 10 min with a 1 ml suspension of RBCs (0.05% hematocrit). After three washes with PBS to remove unbound RBCs, HA-expressing cells had zero to two bound RBC per cell. HA-cells with bound RBC were then incubated in PBS with recombinant La or, in the positive control experiment, with pH 5.0 medium for 5 min. 30 min after La or low pH medium pulse application, we evaluated hemifusion as appearance of HA- cells that acquired PKH26 and fusion as appearance of HA-cells that acquired both PKH26 and CF[85].

**C2C12 cells.** Analysis of fusion of C2C12 cells has been carried out as described earlier[85]. In brief, C2C12 cells were purchased from American Type Culture Collection and propagated in DMEM (Gibco) containing 10% heat-inactivated bovine growth serum and supplemented with penicillin-streptomycin (1%) at 37 °C and 5% $CO_2$. We labeled C2C12 cells with membrane-permeant cell trackers (green CMFDA cell tracker or orange CMRA Cell Tracker) 48 h after placement in the differentiation medium (DMEM containing 5% horse serum (HS, Thermo-Fisher, Catalogue # 26050088) and antibiotics), and co-plated differently labeled cells in a 1-to-1 ratio. The cells were incubated for one more day in the presence of either IgG control, α-La mAb or recombinant La* and then fixed.

### Constructs & Recombinant proteins & Transfection

Recombinant La 1-408, 1-375, 1-375 Q20A_Y24A_D33I, 1-187 and 188-375 were amplified with primers designed with overlapping sequences and inserted between NdeI/HindIII in the V78 pET28A e. coli expression vector, adding to each a N-terminal 6xhis affinity tag. La 1-408 was expressed as a recombinant protein in E. coli and purified using IMAC columns by SD Biosciences (San Diego, CA). The remaining constructs

were transformed into BL2 (DE3) chemically competent e. coli (Thermo Fisher Scientific) and protein expression was induced with IPTG (Sigma). Cells were lysed with Bugbuster (Sigma), and 6xhis-La proteins were affinity purified using HisPur Cobalt Spin columns (Thermo Fisher Scientific), each according to the manufacturer's instructions. Endotoxin contaminates were depleted from affinity purified 6xhis-La proteins using Pierce high-capacity endotoxin removal columns (Thermo Fisher Scientific), according to the manufacturer's instructions. Proteins were then sterile filtered, aliquoted and kept at −80 °C. Recombinant, biotin tagged actin was purchased from Cytoskeleton Inc. (#AB07-A).

Plasmids were introduced into primary osteoclasts at day 2 of RANKL stimulation via jetPRIME according to manufacturer's instructions (Polyplus Transfection). FLAG-La 1-408, FLAG-La 1-375 and FLAG-La 1-375 Q20A_Y24A_D33I plasmids were a gift of the Maraia Lab (NICHD). Briefly, *SSB* (UniProt P05455) was inserted between HindIII and BamHI the pFLAG-CMV2 vector (Sigma). "Uncleavable" La was produced by taking the FLAG-La 1-408 plasmid and making two point mutations at amino acids D371A and D374A, abrogating the caspase cleavage sites at the C-terminal region of the protein (Emory Integrated Genomics Core). siRNAs were introduced into primary osteoclasts after 1 day of RANKL stimulation via Lipofectamine RNAiMAX according to manufacturer's instructions (Thermo Fisher Scientific). Non-targeted (Cat#4390843), *SSB*-targeted (Cat#4392420_ID:s13469) and *Anx A5* (Cat#4390824_ID:s1393) siRNA were introduced at a concentration of 5 ng/ml (Silencer Select, Ambion).

### Antibodies

We used α-Cyclophilin B (Cell Signaling Technology, D1VdJ Rabbit mAb #43603), α-GAPDH (Cell Signaling Technology, D16H11), α-Tubulin (Abcam, 7750), α-RANK (Abcam, 13918), and α-Anx A5 (Abcam, 54775), α-Anx A1 (Abcam, 47661), α-Anx A4 (Abcam, 65846), α-6xHis (Abcam, 18184), α-Actin (Abcam, #11003), α-Syn 1 (Bioss, BS-2962R), α-ANO6 (Invitrogen, #PA5-58610), α-DC STAMP (Millipore, #mabf39-i), and α-FISH (Abcam, 118575), all validated by the manufacturers. We also used control rabbit polyclonal IgG (Abcam, 27478), and IgG2a (Abcam, 18415) was used as an isotype control for α-La, Abcam, 75927).

The key conclusions of the paper are supported by Western blot analysis, fluorescence microscopy, fusion and bone resorption assays using monoclonal murine α-La antibody (α-La mAb; Abcam, 75927). The specificity of this mAb has been validated by the following findings: (1) a drastic increase in La content between monocytes treated with M-CSF and RANKL vs. those treated only with M-CSF detected by mass spectroscopy analysis results in drastic increase in La bands in Westerns and in La staining in immunofluorescence detected with this Ab (Fig. S1 and Fig. 1). (2) siRNA suppression of La expression lowers La staining with this mAb (Fig. 2, Fig. S4d). (3) the presence of La in protein complexes isolated by immunoprecipitation with this mAb has been confirmed with another La Ab (rabbit Anti-SSB antibody Invitrogen, PA5-29763, referred to as α-La rAb) (Fig. S6a). In addition to α-La mAb and α-La rAb, in some experiments we also used rabbit anti-La Phospho-Ser366 antibody (Abcam, 61800, referred to as α-p366 La rAb) that specifically recognizes phosphorylated human La (phosphoSer366). The specificities of La recognition in immunofluorescence microscopy for α-La rAb and α-p366 La rAb were mutually validated by showing the same nuclear staining at the late stage of osteoclast formation (5 days after RANKL application), when phosphorylated La returns to nuclei[33]. The specificity of α-La rAb in Westerns has been confirmed by the experiments, in which we suppressed La expression with siRNA (Fig. S4b).

### Biochemical approaches

Cells were lysed on ice via pulse sonication and rotated end over end at 4 °C for 45 min in the presence of protease inhibitors (cOmplete, Sigma, #118361700010). Steady-state protein levels were evaluated via

SDS-PAGE followed by immunoblotting. The antibodies described were used at the following dilutions for immunoblotting: α-La mAb 1:1000, α-La rAb 1:500, α-Cyclophilin B 1:2000, α-Actin 1:500, α-RANK 1:500, α-Anx A5 1:1000, α-6xHis 1:500, α-Syn 1 1:250, α-ANO6 1:1000, α-DC STAMP 1:500, α-Anx A1 1:250, α-Anx A4 1:250. For secondary antibodies, we used α-Rb Abcam, 7091 or α-Ms Abcam, 7069 1:3000. The selective enrichment of cytosolic vs membrane associated protein fractions was carried out using Mem-PER™ Plus Membrane Protein Extraction Kit (Thermo Fisher Scientific, catalogue # 89842) according to the manufacturer's instructions. Immunoprecipitations were performed as described in[86]. Briefly, multi-protein complexes were sub-stoichiometrically crosslinked using the non-membrane permeable, 12 Å length, cleavable crosslinker 3,3' Dithio-bis(sulfosuccinimidylpropionate) (DTSSP) according to manufacturer's instructions (Thermo Fisher Scientific). Supermolecular complexes were immunoprecipitated using Sheep α-Ms IgG magnetic Dynabeads (Invitrogen) decorated with antibodies targeting proteins of interest (α-La, Abcam, 75927 or IgG2a, Abcam, 18415, 5 μg/ml). Supramolecular complexes were denatured, crosslinking was cleaved via addition of reducing reagents (BME, BioRad) and proteins within these complexes were separated via PAGE. Proteins were transferred and probed for proteins of interest using immunoblotting (as described above). Rb antibodies were used to probe membranes for proteins of interest (α-La PA5-29763, α-Anx A5 14196, α-Anx A1 47661, or α-Anx A4 65846).

**Mass Spectrometry.** 2 biological replicates representing cells from 2 separate donors were treated with either M-CSF alone for 6 days (referred to as M-CSF samples) or M-CSF for 6 days followed by M-CSF and RANKL for an additional 3 days (referred to as RANKL samples). In addition to these 4 biological replicates (2 M-CSF + 2 RANKL samples), we also analyzed 2 additional technical replicates of RANKL samples from donor #2 to help differentiate true "hits" from non-specific proteins.

Bulk proteins were evaluated via SDS-PAGE followed by silver stain (SilverQuest, Thermo Fisher Scientific). Bands of interest were cut from silver-stained gels, distained, subjected to in-gel tryptic digest and evaluated by liquid chromatography coupled with tandem mass spectrometry (Proteomics Core, NHLBI). Mass spectrometry analysis was performed using a Orbitrap Fusion Lumos mass spectrometer (Thermo Scientific) coupled to a Dionex UltiMate 3000-nLC (Thermo Scientific) liquid chromatography system. The data were searched using Sequest within Proteome Discoverer software v1.4 (Thermo Fisher Scientific) against the UniProt human database. The results were filtered with a 1% false discovery rate at the level of proteins and peptides. Raw files were analyzed using the Mascot search engine (v2.3) with criteria: Database, Swiss-Prot (Swiss Institute of Bioinformatics); Taxonomy, eukaryote: Enzyme, trypsin; miscleavages, 2; Variable Modifications, oxidation (M), deamidation (NQ); Fixed Modifications, carbamidomethyl (C); MS peptide tolerance 10 ppm; MS/MS Tolerance, 0.02 Da. The peptide confidence false discovery rates were set to less than 1%.

Out of 131 unique proteins identified in our RANKL samples, 23 were identified consistently across all RANKL replicates. Of these 23, 8 were identified by at least 2 unique peptides and at least 2 PSM in each RANKL sample constituting the threshold of what we consider a bona fide "hit". When compared to the M-CSF samples, 6 of these hits were at least 2-fold enriched (PSMs) in the RANKL conditions. Of these 6 RANKL enriched hits, 4 were actin and keratin isoforms that are known to commonly contaminate mass spectrometry samples. The final 2 hits were Lupus La protein and an isoform of Macrophage-capping protein. Because Macrophage-capping protein is a common actin binding protein in the primary cells we work with, has a significantly lower molecular weight compared to the band we typically saw in our silver gels and considering that we had a high level of actin contaminating

our mass spectrometry samples, we disregarded this hit and moved forward investigating Lupus La protein. For a complete list of the proteins identified in our mass spectrometry data, an annotated "hits" list that includes a list of proteins identified across all RANKL conditions, and a third biological replicate (M-CSF and RANKL samples taken from cells donated by a third individual) where we confirmed our "hits" from the first 2 biological replicates please see Supplementary Dataset 1.

## Transcript Analysis

For real-time PCR, total RNA was collected from cell lysates using PureLink RNA kit following the manufacturer's instructions (Invitrogen # 12183018 A). cDNA was generated from total RNA via reverse transcription reaction using a High-Capacity RNA-to-cDNA kit according to the manufacturer's instructions (Applied Biosystems, # 4387406). cDNA was then amplified using the iQ SYBR Green Supermix (Biorad). All primers were predesigned KiCqStart SYBR Green primers with the highest rank score specific for the gene of interest or GAPDH control and were used according to the manufacturer's instructions (Sigma). All Real-time PCR reactions were performed and analyzed on a CFX96 real-time system (Biorad), using GAPDH as an internal control. Fold-change of gene expression was determined using the $\Delta\Delta Ct$ method[87]. 3–4 independent experiments were performed, and each was analyzed in duplicate.

## Fusion Assays

**Osteoclast fusion.** Osteoclast fusion in our various culture systems was evaluated by fluorescence microscopy[18]. Briefly, cells were fixed with 4% paraformaldehyde at timepoints of interest, permeabilized with 0.1% Triton X-100 and blocked with 5% FBS. Cells were then stained with phallodin-Alexa488 and Hoechst to label cells' actin cytoskeleton and nuclei, respectively. 16 randomly selected fields of view were imaged using Alexa488, Hoechst and phase contrast compatible filter sets (BioTek) on a Lionheart FX microscope using a 10x/0.3 NA Plan Fluorite WD objective lense (BioTek) using Gen3.10 software (BioTek). Osteoclast fusion efficiency was evaluated as the number of fusion events between cells in these images[88]. In brief, since regardless of the sequence of fusion events, the number of cell-to-cell fusion events required to generate syncytium with N nuclei is always equal to N-1, we calculated the fusion number index as $\Sigma (N_i - 1) = N_{total} - N_{syn}$, where $N_i$ = the number of nuclei in individual syncytia and $N_{syn}$ = the total number of syncytia. We normalized the number of fusion events to the total number of nuclei (including unfused cells) to control for small variations in cell density from dish to dish. In contrast to traditional fusion index measurements, this approach gives equal consideration to fusion between two mononucleated cells, one mononucleated cell and one multinucleated cell and two multinucleated cells. In traditional fusion index calculations, fusion between two multinucleated cells does not change the percentage of nuclei in syncytia. If instead one counts the number of syncytia, a fusion event between two multinucleated is not just missed but decreases the number of syncytia. In contrast, the fusion number index is inclusive of all fusion events.

**Osteoclast Membrane Fusion Synchronization.** Osteoclast fusion was synchronized as described in[18,89]. Briefly, osteoclast media was refreshed with 100 ng/ml M-CSF, 100 ng/ml RANKL and 350 μm lauroyl-LPC 72 h post RANKL treatment. Following 16 h, LPC was removed via 5 washes with fresh media and cells were allowed to fuse in the presence or absence of antibody treatment or recombinant La at the concentrations described in the figure legends for 90 min.

**HA0-RBC fusion assay.** To test whether La is capable of mediating fusion, we applied this protein to HA0-expressing cells with RBCs tightly bound by the interactions between sialic acid receptors at the surface of RBCs and HA1 subunit of HA0[90]. HA0, an uncleaved form of HA, mediates binding but does not mediate fusion. HA0-cells were twice washed with PBS and incubated for 10 min with a 1 ml suspension of RBCs (0.05% hematocrit). We washed HA-cells with zero to two bound RBC per cell with PBS to remove unbound RBC and then applied 40 nM FL La. Fusogenic activity was assayed by fluorescence microscopy 1 h after La application as the ratio of the numbers of lipid probe (PKH26)-labeled HA-cells, respectively, to the total number of HA-cells. In positive control experiment we treated HA-cells with 1 μg/ml trypsin for 5 min at room temperature. Then the cells were exposed for 5 min to the pH 5.0 medium at the room temperature. Images were taken 30 min later.

**Myoblast fusion assay.** We scored myoblast fusion as described in[85]. We stained the nuclei with Hoechst (10 mg/ml stock diluted 1,000-fold for 30 min at the room temperature) and took the images on fluorescence microscope using appropriate excitation and emission filters and MicroManager v.1.4.23 Open Source Microscopy Software for image acquisition. We also used Fiji/ImageJ open-source image processing package v.2.1.0/1.53c for viewing and scoring. Fusion completion was detected as formation of multinucleated cells (defined as cells with more than 1 nucleus per cell) and quantified as the percentage of cell nuclei present in multinucleated cells normalized to the total number of cell nuclei. For each condition, ≥10 randomly chosen fields of view were imaged per condition per experiment.

## Liposome experiments

**Liposome binding assay.** Multilamellar liposomes were formed from pure PC or 9:1 (mole/mole) mixture of PC and PS. Both lipid compositions were supplemented with 0.5 mol % of lissamine rhodamine phosphatidylethanolamine. To prepare liposomes, lipid stock in benzene/methanol (95:5) was frozen in liquid nitrogen and freeze-dried overnight using SpeedVac (Savant). Dried lipid was resuspended in aqueous buffer (100 mM NaCl, 10 mM Hepes, pH 7.0) at 1 mM total lipid concentration and vortexed. Proteins (67.5 nM of A5 and 67.5 nM of La) and 2 mM $CaCl_2$ were added to the liposomes (total lipid concentration 0.5 mM) and the mixtures were incubated on ice for 30 min and a small volume was collected as a "non-centrifugated sample". To pellet, liposomes were centrifugated at 15,000 g for 20 min. Based on rhodamine fluorescence, >90% of liposomes were pelleted across all conditions regardless of lipid content or the addition of proteins or $Ca^{2+}$. Centrifugated samples were then fractionated into a top, liposome-depleted fraction and a bottom fraction containing liposomes and liposome-bound proteins. Fractions were then solubilized via addition of Laemmli buffer (BioRad) and separated via SDS-PAGE, as described above. Recombinant La and recombinant Anx A5 were detected via their n-terminal 6xHis tag via α-6xHis antibody (Abcam) and signals for soluble vs liposome bound protein fractions were evaluated via densitometry. Data were presented as a percentage of protein signal bound to liposomes where $Liposome\ Bound\ Protein = \frac{Liposome\ Fraction - Soluble\ Fraction}{Liposome\ Fraction + Soluble\ Fraction}$.

In this presentation, liposome bound protein represents 100% of the total protein, if all the protein is in the liposome fraction, and 0%, if the protein is equally distributed between the liposome and soluble fractions.

**Lipid Mixing Assay.** Large unilamellar vesicles were prepared according to protocol described in[85]. In short, 10 mM lipid stock in benzene: methanol = 95:5 was freeze dried overnight; lipid powder was resuspended in aqueous buffer (100 mM NaCl, 10 mM Hepes, pH 7.0) at 0.5 mM final total lipid concentration; lipid suspension was freeze-thawed 10 times by repeated submersion into liquid nitrogen and water bath and then extruded 10 times through 2 stacked polycarbonate membrane with nominal pore size of 200 nm. Unlabeled

# Article

liposomes were prepared from 1:1 mixture of DOPC and DOPS. Fluorescently labelled liposomes contained 0.25 mol % of TopFluor PE and 0.5 mol % rhodamine PE. Release of FRET upon lipid mixing was followed as dequenching of TopFluor fluorescence measured at 510 nm with excitation at 480 nm. Addition of 13.5 nM Anx A5, 13.5 nM La and 2 mM $Ca^{2+}$ to 1:10 mixture of labeled and unlabeled liposomes (total lipid concentration of 13.75 μM) had no effect on TopFluor fluorescence, while addition of only 5 mM $Ca^{2+}$ in control experiments induced efficient lipid mixing.

## Treatments with A01 and EGTA

To suppress PS externalization mediated by TMEM16 scramblases, monocyte-derived osteoclasts 4 days after RANKL application were treated with 60 μM A01 in the complete medium for 1 h at 37 °C. The cells were washed 3 times with PBS and fixed, as described below for immunofluorescence analysis.

To disrupt $Ca^{2+}$ dependent binding of endogenous Anx A5 to PS at the surface of the monocyte-derived osteoclasts 4 days after RANKL application, we incubated the cells with 10 mM buffered EGTA in the complete medium for 10 min at 37 °C. After 3 washes with PBS the cells were fixed for the analysis.

## Recombinant La Membrane Binding

Recombinant La 1-375 and BSA were covalently labeled with CF555 or CF488A dyes, respectively, following manufacturer's instructions (Biotium). Labeled La and BSA were added to fusing RAW 264.7 osteoclasts or C2C12 myoblasts at a final concentration 1.67 μg/ml and incubated at 37 °C for 1 hr. Cells were subsequently washed 4 times with fresh media and fixed for analysis.

## Fluorescence Microscopy Imaging

In the immunofluorescence experiments we washed the cells with PBS and fixed with warm freshly prepared 4% formaldehyde in PBS (Sigma, F1268) at 37 °C. The cells were washed with PBS. To permeabilize the cells, we incubated them for 5 min in 0.1% Triton X100 in PBS. The cells were washed three times with PBS and placed into PBS with 10% FBS, for 10 min at the room temperature to suppress non-specific binding. Then the cells were incubated with primary antibodies for 1 h in PBS with 10% FBS. The antibodies α-La mAb, α-Fish, α-Anx A5, α-p366 La and α-RANK were each used at 1:100 for immunostaining. After 5 washes in PBS, we placed the cells for 1 h at room temperature in PBS with 10% FBS with secondary antibodies (either Anti-rabbit IgG Fab2 Alexa Fluor or Anti-mouse IgG Fab2 Alexa Flour ® 488, both Cell Signaling Technology, Catalogue # 647 4414 S and # 4408 S, respectively, in 1:500 dilution) and then again washed the cells 5 times with PBS.

In the experiments that required immunostaining of non-permeabilized cells (Fig. 3b, c), we first fixed the cells as above. We washed the cells with PBS (without $Ca^{2+}$/$Mg^{2+}$) and placed them into PBS with 10% FBS, for 10 min at the room temperature to suppress non-specific binding. Then the cells were incubated with primary antibodies (for La, α-La mAb, 1:100 dilution in PBS with 10% FBS and for Anx A5, 1:100 dilution) for 1 h at the room temperature. After 5 washes with PBS, the cells were placed into PBS with 10% FBS, for 10 min at the room temperature. Then the cells were incubated with secondary antibodies, as described above (1 h in PBS with 10% FBS at room temperature) and, finally, washed 5 times with PBS.

Images were captured on a Zeiss LSM 800, confocal microscope using a C-Apochromat 63x/1.2 water immersion objective lens.

## Bone Resorption

Bone resorption was evaluated using bone resorption assay kits from Cosmo Bio USA according to the manufacturer's instructions. In short, fluoresceinamine-labeled chondroitin sulfate was used to label 24-well, calcium phosphate-coated plates. Human, monocyte-derived osteoclasts were differentiated as described above, using alpha MEM without phenol red. Media were collected at 4–5 days post RANKL addition, and fluorescence intensity within the media was evaluated as recommended by the manufacture.

## Statistics and reproducibility

Each graph presents data from three independent experiments unless stated otherwise in the legend. Data were assembled and analyzed using GraphPad Prism 8.0. For each experiment, cells from the same passage, donor or animal were paired across the differing conditions described and assessed using a paired t test. All functional dependencies reported were observed in each independent experiment. However, as known for the human monocyte-derived osteoclasts 10, time courses of osteoclastogenic differentiation and baseline extents of fusion considerably varied for monocytes from different donors. Moreover, in experiments where we transfect or otherwise treat these cells at the same timepoint, the extent of transfection can differ based on their differing rates of differentiation and baseline extents of fusion. For some fusion experiments where consistent results were observed but with considerable verbality, we analyzed statistical significance using a ratio paired t test, where raw values are logarithmically transformed and then assessed. In our analysis of the HA0-RBC experiments, Wilson-method-based confidence limits for binominal proportion were calculated in R (v. 4.1.1) using binconf function of Hmisc package (v. 4.5.0). All experiments presented as representative micrographs or gels were repeated at least 3 times with similar results.

## Reporting summary

Further information on research design is available in the Nature Portfolio Reporting Summary linked to this article.

## Data availability

Source data are provided with this paper.

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

## Acknowledgements

We thank Drs Marjan Gucek, Yanling Yang and the proteomics core at the National Heart, Lung and Blood Institute for their assistance in identifying La protein. We thank the National Institutes of Health Department of Transfusion Medicine for isolating the monocytes used in this study. Dr. Joshua Zimmerberg, NICHD, NIH is acknowledged for the kind gift of hemagglutinin expressing cells. The research in L.V.C.' and R.J.M.' laboratories was supported by the Intramural Research Program of the Eunice Kennedy Shriver National Institute of Child Health and Human Development, National Institutes of Health. The research in M.T.C.' laboratory was supported by the Intramural Research Program of the National Institute of Dental and Craniofacial Research, National Institutes of Health. The research reported in this publication was also supported by the of Research on Women's Health (ORWH) through the Bench to Bedside Program award # 884515 to L.V.C. and M.T.C.

## Author contributions

J.M.W., E.L., and L.V.C. designed osteoclast fusion experiments. J.M.W. and E.L. performed these experiments and analyzed the data. J.M.W. has designed, performed and analyzed all biochemical experiments. J.M.W., L.F.D.C, M.T.C. and LV.C. have designed bone marrow explant experiments and J.M.W. has performed them and analyzed the data. Liposome experiments were designed, carried out and analyzed by K.M. and J.M.W. Immunofluorescence experiments, experiments with hemagglutinin expressing cells and myoblasts have been designed, performed and analyzed by E.L. and J.M.W., S.M and R.J.M. have contributed to the work an unpublished triple mutant of La, which S.M. has characterized and advised J.M.W. in the use of this mutant. R.J.M. and M.T.C. have

contributed to the design of the work and interpretation of the data within the context of La protein biology and bone biology, respectively. J.M.W. and L.V.C. wrote the manuscript with assistance from all authors.

## Funding

## Competing interests
The authors have no competing interests apart from a pending patent application (U.S. Patent Application No. 63/155,896 "La protein as a novel regulator of osteoclastogenesis" with LVC, EL, JMW, listed as inventors, with patent held by NICHD, NIH).
