## [Peer Review File · Nature Communications]

CELL SURFACE-BOUND LA PROTEIN REGULATES THE CELL FUSION STAGE OF OSTEOCLASTOGENESISReviewer #1 (Remarks to the Author):

In this manuscript, the authors proposed that cell surface protein La may control osteoclast fusion.

The detailed understanding of the molecular mechanisms underlying osteoclast fusion is important issue in the bone biology field, and the authors' hypothesis on the role of La in osteoclast fusion is interesting. However, this manuscript has several critical concerns. Importantly, the authors failed to provide convincing evidence for the importance of La in osteoclast fusion in vivo. The molecular mechanism underlying La-mediated osteoclast fusion is also unclear.

Taken together, this manuscript is not suitable for publication in Nature Communications in its current form.

Major concerns

1. The authors should generate and analyze osteoclast-specific La deficient mice (e.g. *Ssb flox/flox Ctsk-Cre*) to prove the physiological relevance of La in osteoclast fusion in vivo.
2. The molecular mechanism underlying how La facilitates osteoclast fusion is unclear. Although the authors proposed that La may interact with Anx A5, there is no direct evidence showing the importance of interaction between La and Anx A5 in osteoclast fusion. The authors should show that the effect of La is dependent on Anx A5 by using Anx A5 deficient cells, and should clarify how the La/Anx A5 complex promotes osteoclast fusion.
3. In Fig.S4, the authors should check the effect of RNAi suppression of La on the expression levels of other key fusogenic factors such as DC-STAMP, OC-STAM, ATP6v0d2 and *Sbno2*.
4. Given that La is ubiquitously expressed in various cell types, it is interesting to know whether La functions as a fusogenic factor only in osteoclasts. The authors should check the effect of La deletion on myocyte fusion and discuss molecular mechanisms underlying the cell type-dependent functions of La protein.

Reviewer #2 (Remarks to the Author):

Review of Whitlock et al., for Nature Communications 2022

The manuscript by Whitlock and colleagues describes the discovery of a significant novel function of the mammalian La protein. The mammalian full length La protein has previously been well characterized as an RNA binding protein with roles in the processing and metabolism of a variety of RNAs in the nucleus and cytoplasm. Through a series of experiments, the authors elucidated that a protease-cleaved form of La has an independent function at a different time and in a different location (on the cell surface) where it acts as an important regulator of osteoclast membrane fusion. They also discuss that the shorter form of La could potentially be a significant new therapeutic target for diseases that involve bone remodeling, for example, osteoporosis. The protein is also an interesting example of a protein with two very different functions in different cellular localizations, which would be of interest to many labs interested in understanding and predicting protein functions and regulation in general. Although I do not personally have experience in some of the methods that were used, in my opinion the data analysis, interpretation, and conclusions appear to be accurate and described with enough detail to be reproduced.

My suggestions are mainly grammatical:

- p. 3 – traffics -> trafficks as verb
- p. 10 – 2nd paragraph – osteoclasts misspelled
- p. 12 first line, add word "but" after comma
- p.12 3rd paragraph – concentrations of RANKL
- p. 15 middle of 2nd paragraph – Anx A1, which Anx A4, which

- p. 15 protein complexes further support the
- p. 17 2nd paragraph – low in osteoclast precursors, precursors maintain very low levels = repetitive
- p. 19 2nd line - targets instead of approaches
- Various places throughout – commas needed to split sentence in 2 after “and”, other places “the” or “a” sometimes need to be added
- Methods – E in E. coli should be capitalized, Thermo should probably be ThermoFisher,
- p. 5 in methods – sterile is misspelled as steril
- Antibodies section – please check positions of parentheses, check for “a” where it should be the letter alpha in a few places (p. 6)
- p. 7 stoichiometrically is misspelled
- p. 10 Alexa Fluor is misspelled

Reviewer #3 (Remarks to the Author):

The authors report that the spatial-temporal regulation of La protein contributes to monocyte-to-osteoclast differentiation. While La protein is typically thought to play roles in RNA metabolism, the manuscript describes a new role of cell-surface La protein in osteoclast formation. Upon phosphorylation at S366 and caspase cleavage, a low molecular weight species of La is present at the cell surface. Proteomics identified that the levels of this low-molecular weight species of La correlates with multinucleated osteoclasts. This novel La function appears to be independent of the canonical N-terminal binding domains of La proteins. Through biochemical and cellular approaches, the authors show that fusion can be regulated by the levels of La. Moreover, the authors show an ex vivo model of osteoclast formation in fibrous dysplasia driven by mutations in Gas. With this model, the manuscript shows that the application of anti-La antibodies can reduce osteoclast fusion, suggesting a possible use of La protein as a potential therapeutic target. The findings reported by the manuscript authors are noteworthy and of interest to the field. More information or clarification should be added to ensure the data is presented clearly.

Major Comments:

- 1)“Since La, on its own, initiates neither hemifusion nor fusion between bound membranes, it is unlikely that La works as a “blue collar worker” directly catalyzing and/or driving membrane fusion. More likely La, as a “white collar worker”, recruits or stimulates other components of the osteoclast fusion complex.”
 - The authors are inappropriately referring to proteins in a manner that perpetuates inequities in our field. Referring to proteins as “blue collar” and “white collar” is unnecessary and vague. Rather they should be referred to as direct or indirect actions.
- 2)The use of technical replicates and biological replicates is unclear in the manuscript, along with the use of statistics
 - Figure 1b. SEM is represented as bars in figure 1b and can be misleading. It is better to represent data as a range on the graph. There are no asterisks visible in the figure, but a sentence regarding the symbol and the statistical test is present in the figure legend.
 - Figure S2, Figure S5d. Bars (n=2) are represented on the graph without a description of what they refer to. Readers may assume these are SEM or SD values. The authors should clearly indicate that these are ranges.
 - In Figure 4, Figure 3g: statistics are reported for two replicates when one cannot determine a normal distribution from two measurements.

Minor Comments:

- 1)The manuscript stressed the N-terminal RNA binding domains of La, naming the triple mutant (Δ RNA). To confirm that the function of La in this context is RNA-independent, the IP pull downs with La should be performed with RNAses. The authors should discuss the C-terminal RRM included in their constructs and how it may play novel La-RNA binding roles in their model. It is difficult to determine that the role of La in osteogenesis is independent of RNA as the protein may have different RNA binding residues presented on the cell surface.
- 2)Either figure S1d incorrectly refers back to Figure 1b, or the connection between the data is unclear.
- 3)Figure S1 panel lettering is off.
- 4)Figure 7c is not called out in the text.
- 5)The use of a period after Doxy. makes reading the manuscript difficult. I suggest using Doxy as

a substitute.

6) It is unclear how many technical and biological replicates were used in the tandem mass-spec data. The table representing six samples (S1d should be b, see point 3) and the description in the figure legend is confusing. It appears the samples were taken from two biological replicates with three technical replicates each. The language should be clarified. Additional information should be added to the methods as to how the mass-spec data was filtered.

7) "La's tight regulation during osteoclastogenesis must be carried out at the protein level, as, despite the scarcity of La protein, M-CSF derived precursors contain even more La transcript (gene SSB) than after RANKL application (Fig. S1e)."

• The authors have shown that osteoclastogenesis relies on the appropriate post-translational species and trafficking of La-protein. However, they cannot conclude that other forms of post-transcriptional regulation contribute more to these events without the appropriate controls. I suggest softening the language for this sentence.

Reviewer #4 (Remarks to the Author):

The manuscript by Whitlock et al describes the involvement of La protein (SSB) in osteoclast fusion. Osteoclasts are essential for skeletal remodelling due to bone-resorbing properties, and derive from monocytes upon stimulation with cytokines including M-CSF and RANKL. By monitoring changes in protein levels during monocyte differentiation *in vitro*, the authors discover an involvement of the RNA binding protein La, which they aim to characterise in the rest of the study. Although understanding the molecular mechanism of regulated osteoclast fusion is a very important outstanding question in the field, the current study lacks clarity, fails to present a convincing mechanism of La-protein action and moreover suffers in part from poor experimental design and lack of experimental controls. Thus, the present study and conclusions drawn seem rather premature and add confusion instead of offering new insights into the biology of osteoclast fusion. Moreover, the manuscript has not been assembled with sufficient care (e.g. an antibody annotation conflict Fig 1D, and incorrect in-text references to Fig3).

Major concerns

1) A more unbiased quantitative approach seems warranted to monitor proteomic changes during osteoclast differentiation/fusion.

2) Given the central role of different molecular species of La protein, which the authors claim to detect by using "specific" antibodies, the antibodies in question must be validated more carefully. The fact that La protein exhibits a different MW on blots presented in figures 1 and 2 (inconsistent use of MW standards) is very confusing. Furthermore, it is unclear if the blot in 1d is for La protein or rather RANK (Abcam 13918) as stated in the figure legend. Figure 1d and 1e are supposed to be blotted for La with the same antibody, those should be run on the same gel/blot in order to give a clear idea of steady-state FL and cleaved La species during the course of osteoclast formation. In order to specifically detect FL La, the authors use a phospho-specific antibody (abcam 61800), but fail to provide evidence (on a WB) that this indeed detects specifically the full-length protein. Thus all claims made in this regard seem highly questionable. More surprisingly the authors claim to detect (the cleaved form of) La at the cell surface. However, given the absence of a previously reported or experimentally addressed mechanism of La-protein secretion as well as the strikingly different cell surface staining pattern in human and mouse cells (Fig 3d-c) better specificity controls are required (e.g. does RNAi abolish/reduce staining).

Also the Antibody PA5-29763 (which is referred to as a-LMW La) is actually raised against AA 2-242 of human SSB, and so should detect both the long and short form of the protein.

3) In order to evaluate the more canonical role of RNA-binding protein La in post-transcriptional regulation of mRNAs, the authors evaluate mRNA levels of known osteoclast fusion factors. However, the literature and even the publication cited by the authors (Sommer et al 2011) mainly describes a role of La in mRNA translation. Thus, the authors must look at protein and not mRNA levels to reach a valid conclusion.

4) In figure 2, the fusion promoting effect of ectopic La expression should best be evaluated in the RNAi background. This type of rescue experiment would additionally validate RNAi specificity. Additionally, the authors have to provide WB analysis of different constructs. mRNA/RT-qPCR is

insufficient and likely flawed by the large excess of input/transfected DNA.

5) As mentioned above the cell surface staining with anti-La presented in Figure 3 requires further validation. Furthermore, the use of abs to inhibit fusion has to be controlled better. It is not clear that the observed effect is not due to high (background) binding by anti-La, where control abs do not bind, or bind to a much lesser extent.

6) It remains unclear if the fusion stimulating effect of recombinant La (Fig.4) is exerted on the cell surface and not inside the cell. Certainly, the prolonged (overnight) incubation time with recombinant protein would allow for cellular uptake and a cytosolic effect. Thus, further control experiments are warranted.

7) The authors have addressed whether La acts itself as a fusogen. This is described only in the text but the underlying data is not shown/has been omitted?

8) In order to evaluate a role for La as a fusion regulator the authors describe its interaction with Annexin A5 (a protein previously reported by the same lab to be involved in fusion). However, data presented in Fig S5 are not convincing as lysate controls are missing or not run properly alongside the immuno-precipitated samples. The data representing liposome binding experiments (FigS5 D) are confusing in terms of what has been tested but essential negative controls seem to be missing. Also, the primary data (protein gels should be shown). More importantly however, the authors fail to provide any evidence that would help to answer the question of how the postulated interaction with Annexin A5 would stimulate osteoclast fusion.

9) Given insufficiently controlled RNAi and anti-La surface binding/inhibition of fusion experiments, effects observed by the authors on bone resorption (Fig 5) in a osteoblast/osteoclast co-culture setting (Fig 6) and in a mouse model exhibiting an excess of osteoclast fusion (Fig 7), do not really add to the story, as unfortunately they cannot be interpreted with sufficient confidence.

Re: NCOMMS-21-47862-T

Point-by-Point Response

REVIEWER 1.

We greatly appreciate the critiques and suggestions that helped us to strengthen the manuscript.

Comment 1. “Importantly, the authors failed to provide convincing evidence for the importance of La in osteoclast fusion in vivo. The molecular mechanism underlying La-mediated osteoclast fusion is also unclear.”

The authors should generate and analyze osteoclast-specific La deficient mice (e.g. *Ssb* flox/flox *Ctsk*-Cre) to prove the physiological relevance of La in osteoclast fusion in vivo.

We agree with the Reviewer that it is important to test whether La is indispensable for osteoclast formation. The challenge here is the essential role of La as an RNA chaperone. As noted in our manuscript, “...La is required for mouse development and the establishment of embryonic stem cells³⁸, and efforts to specifically delete La in B cell progenitors and the forebrain have resulted in the loss of these cell types³⁷.” Our preliminary data suggest that La is also required for survival of the cells of monocyte lineage. We collected bone marrow cells from mice with a conditional La allele (Gaidamakov et al., *Mol Cell Biol.* 2014, 34, 123), incubated these cells with M-CSF to generate osteoclast precursors and transduced them to express Cre recombinase to remove La. Sadly, we could not answer the question of whether La is essential for osteoclast fusion because all the cells consistently die. Note that transduction with GFP-expressing virus alone had no effect on cell viability. We hope our on-going work on identification of regions within La that are critical for its role in OC fusion and dispensable for La’s role as an RNA chaperon will allow us to generate mice, in which full-length La in cells of osteoclast lineage will be replaced with La engineered to retain RNA-binding sequences but lose its ability to promote OC formation. We consider this project to be beyond the scope of the present paper.

2. The molecular mechanism underlying how La facilitates osteoclast fusion is unclear. Although the authors proposed that La may interact with Anx A5, there is no direct evidence showing the importance of interaction between La and Anx A5 in osteoclast fusion. The authors should show that the effect of La is dependent on Anx A5 by using Anx A5 deficient cells, and should clarify how the La/Anx A5 complex promotes osteoclast fusion.

As suggested by the Reviewer, we have deepened our analysis of the La-Anx A5 interactions. In new experiments we confirmed direct La-Anx A5 binding by streptavidin pulldown of 6xHis-La from mixtures of recombinant Biotin-AnxA5 and 6xHis-La (Fig. 5b). Since Anx A5 deficiency and reagents targeting cell surface Anx A5 are already known to suppress osteoclast fusion (Verma et al. the JBC, 2018), we focused on examining whether La association with the cell surface depends on Anx A5. Anx A5 binding to membranes is known to depend on Ca²⁺ and phosphatidylserine (PS). Our data in liposome assay showed that La binding to liposomes depends on Anx A5, Ca²⁺ and PS. In new experiments we found that lowering the amounts of cell surface Anx A5 by washing the osteoclast precursors with EGTA to remove Ca²⁺ lowered the amounts of cell surface La (Fig. 5e,f). Lowering the amounts of cell surface bound Anx by suppressing the externalization of PS by an inhibitor of TMEM16 lipid scramblase CaCCinh-A01 (Verma et al. the JBC, 2018), also lowered the amounts of cell surface La (Fig. 5g). Novel and unexpected finding that La

binds to Anx A5 is now supported by following lines of evidence: (1) La-containing protein complexes isolated by immunoprecipitation from osteoclast lysates contain Anx A5; (2) Anx A5-containing protein complexes contain La; (3) La binding to liposomes depends on Anx A5, Ca²⁺ and PS; (4) streptavidin pulldown of 6xHis-La from mixtures of recombinant Biotin-AnxA5 and 6xHis-La; (5, 6) loss of cell surface La observed after lowering the cell surface Anx A5 by either Ca²⁺ removal or suppressing PS cell surface exposure. Our new findings (items 4, 5 and 6 above) provide additional evidence for La-Anx A5 binding, explain how La, a soluble protein, associates with cell membrane, and connect La function in osteoclastogenesis with non-apoptotic phosphatidylserine exposure signaling pathways that regulate osteoclast fusion.

We agree that the mechanism by which La promotes fusion remains to be understood. It could take many years and work of many labs to resolve this mechanism. As an analogy, 9 years after the work of Millay *et al.*, Nature, 2013 showing Myomaker dependence of myoblast fusion and after a number of high impact follow up studies exploring different aspects of myomaker contributions, we still do not know how Myomaker facilitates fusion. Similarly, while the dependence of cell-cell fusion in *C. elegans* on EFF-1 has been identified in Mohler *et al* Dev Cell 2002 followed by a number of high impact papers, mechanistic ideas on how EFF1 works came only in 2014 in Pérez-Vargas *et al.*. While the mechanisms by which La promotes osteoclast fusion remain to be clarified, we do consider our discovery of the regulatory role of La in the cell fusion stage of osteoclastogenesis to be a major advance and finding that osteoclast fusion is regulated by cell-surface complexes of La and Anx A5 bound to phosphatidylserine to be an important mechanistic insight. Moreover, this entirely novel function for a ubiquitous, nuclear RNA chaperone was completely unexpected, and demonstrates that evolution has repurposed La for alternative, non-canonical functions outside the cell. Furthermore, our findings identify easily accessible cell-surface La as an intriguing target for treating bone loss diseases.

3. In Fig.S4, the authors should check the effect of RNAi suppression of La on the expression levels of other key fusogenic factors such as DC-STAMP, OC-STAM, ATP6v0d2 and Sbn02.

The conclusion that RNAi suppression of La inhibits fusion by lowering cell surface La rather than by a loss of other factors is now supported by finding that fusion inhibition by La targeting RNAi is partially rescued by application of recombinant La (Fig. S4e).

In addition, in response to the Reviewer's concern and at the suggestion of Reviewer #4 we have added Figure S4b which assesses how La targeting RNAi impacts the steady-state protein levels of La, Syn1, Anx A5, ANO6 and DC-STAMP. While we see ample reduction of steady-state La protein levels, we observe no changes in any of the other proteins, including the OC fusion related proteins Syn1 or DC-STAMP.

We have not analyzed expression of Sbn02 because this regulator of osteoclast fusion promotes fusion by enhancing the expression of DC-STAMP (Maruyama *et al.*, Strawberry notch homologue 2 regulates osteoclast fusion by enhancing the expression of DC-STAMP, J. Exp. Med. 2013 Vol. 210, 1947-1960). We have not explored the effects of La knockdown on ATP6v0d2 because expression of this protein is controlled by NFATc1 (Kim *et al.*, NFATc1 induces osteoclast fusion via up-regulation of Atp6v0d2 and the dendritic cell-specific transmembrane protein (DC-STAMP), Mol Endocrinol 2008 Jan;22(1):176-85) and NFATc1 expression does not change as a result of La suppression. We did attempt to assess OC-STAMP levels using a commercially available, previously published antibody, but

unfortunately this reagent did not produce any signal in control or La suppressed conditions, leaving us unable to comment on OC-STAMP. We do not consider adding more proteins to the list of tested as critical for the paper because our data show rapid inhibition and promotion of synchronized fusion by antibodies and recombinant proteins added to already differentiated osteoclast precursors. These findings show that La functions downstream of pre-fusion differentiation steps.

4. Given that La is ubiquitously expressed in various cell types, it is interesting to know whether La functions as a fusogenic factor only in osteoclasts. The authors should check the effect of La deletion on myocyte fusion and discuss molecular mechanisms underlying the cell type-dependent functions of La protein.

In revision we have added new data suggesting that mechanisms underlying cell surface La promotion of osteoclast fusion are not shared in myoblast fusion (Fig. S7c,d). Specifically, we found that application of α -La antibodies (α -La mAb) and recombinant La (La 1-375 La*) (the reagents that inhibited and promoted osteoclast fusion, respectively) to differentiating C2C12 myoblasts have no effect on myoblast fusion.

REVIEWER 2.

We greatly appreciate the suggestions that helped us to strengthen the manuscript.

My suggestions are mainly grammatical:

p. 3 – traffics -> trafficks as verb

p. 10 – 2nd paragraph – osteoclasts misspelled

p. 12 first line, add word “but” after comma

p.12 3rd paragraph – concentrations of RANKL

p. 15 middle of 2nd paragraph – Anx A1, which Anx A4, which

p. 15 protein complexes further support the

p. 17 2nd paragraph – low in osteoclast precursors, precursors maintain very low levels = repetitive

p. 19 2nd line - targets instead of approaches

Various places throughout – commas needed to split sentence in 2 after “and”, other places “the” or “a” sometimes need to be added

Methods – E in E. coli should be capitalized, Thermo should probably be ThermoFisher,

p. 5 in methods – sterile is misspelled as steril

Antibodies section – please check positions of parentheses, check for “a” where it should be the letter alpha in a few places (p. 6)

p. 7 stoichiometrically is misspelled

p. 10 Alexa Fluor is misspelled

Thanks. All suggestions have been addressed.

REVIEWER #3

We greatly appreciate the critiques and suggestions that helped us to strengthen the manuscript.

The authors report that the spatial-temporal regulation of La protein contributes to monocyte-to-osteoclast differentiation. While La protein is typically thought to play roles in

RNA metabolism, the manuscript describes a new role of cell-surface La protein in osteoclast formation. Upon phosphorylation at S366 and caspase cleavage, a low molecular weight species of La is present at the cell surface. Proteomics identified that the levels of this low-molecular weight species of La correlates with multinucleated osteoclasts. This novel La function appears to be independent of the canonical N-terminal binding domains of La proteins. Through biochemical and cellular approaches, the authors show that fusion can be regulated by the levels of La. Moreover, the authors show an ex vivo model of osteoclast formation in fibrous dysplasia driven by mutations in $G\alpha_s$. With this model, the manuscript shows that the application of anti-La antibodies can reduce osteoclast fusion, suggesting a possible use of La protein as a potential therapeutic target.

The findings reported by the manuscript authors are noteworthy and of interest to the field. More information or clarification should be added to ensure the data is presented clearly.

Major Comments:

1) "Since La, on its own, initiates neither hemifusion nor fusion between bound membranes, it is unlikely that La works as a "blue collar worker" directly catalyzing and/or driving membrane fusion. More likely La, as a "white collar worker", recruits or stimulates other components of the osteoclast fusion complex."

- The authors are inappropriately referring to proteins in a manner that perpetuates inequities in our field. Referring to proteins as "blue collar" and "white collar" is unnecessary and vague. Rather they should be referred to as direct or indirect actions.

Thanks, Agreed, we have changed the language as suggested.

2)The use of technical replicates and biological replicates is unclear in the manuscript, along with the use of statistics

- Figure 1b. SEM is represented as bars in figure 1b and can be misleading. It is better to represent data as a range on the graph. There are no asterisks visible in the figure, but a sentence regarding the symbol and the statistical test is present in the figure legend.

Agreed. We understand the Reviewer's concern and have replaced the previous graph with one now representing each replicate at each timepoint with dots. We hope that this will more clearly relates the temporal relationship of RANKL addition and osteoclast formation that we observe, as well as fully illustrates the variability observed between cells donated by different individuals. We have corrected the sentence concerning asterics and statistical tests, please forgive our oversight.

- Figure S2, Figure S5d. Bars (n=2) are represented on the graph without a description of what they refer to. Readers may assume these are SEM or SD values. The authors should clearly indicate that these are ranges.

- In Figure 4, Figure 3g: statistics are reported for two replicates when one cannot determine a normal distribution from two measurements.

We appreciate the Reviewer's critique. Please see where S2>Now S2a, S5d>Now Figure 5d and Figure 3g>Now Supplementary Figure 5b have now been updated. We have removed error bars and now show individual points for each replicate, to avoid the confusion the Reviewer pointed out with our error bars. For Figure 4b, we have kept a bar simply to maintain consistency between the conditions shown, but have clearly noted in the figure

legend that the heat inactivated condition is n=2 and that the bar is an indication of the range, not SEM.

Minor Comments:

1)The manuscript stressed the N-terminal RNA binding domains of La, naming the triple mutant (Δ RNA). To confirm that the function of La in this context is RNA-independent, the IP pull downs with La should be performed with RNases. The authors should discuss the C-terminal RRM included in their constructs and how it may play novel La-RNA binding roles in their model. It is difficult to determine that the role of La in osteogenesis is independent of RNA as the protein may have different RNA binding residues presented on the cell surface.

Thanks! We agree that naming the triple mutant Δ RNA without direct analysis of RNA binding has been inappropriate. In the revised MS, we have renamed this mutant into La*. We agree that we cannot exclude a novel role of RRM2 bound RNA in the structure and function of cell surface LA and we added the corresponding comment to our discussion. “Since recent studies found RNAs present on the surface of living cells that are involved in monocyte interactions ^{62, 63}, it remains possible that RNA is involved in surface La’s role in regulating osteoclast fusion. However, even if the function of La in osteoclast formation depends on La-RNA interactions at the cell surface through some yet unknown mechanism, possibly involving RRM2 ³⁰, this novel function fundamentally differs from the classical functions of La dependent on its La domain- and RRM1- mediated RNA binding in the nucleus and cytoplasm. “

2)Either figure S1d incorrectly refers back to Figure 1b, or the connection between the data is unclear.

We thank the Reviewer for drawing our attention to an incorrect reference in our supplementary figure. This has been corrected so that the supplementary figure now refers to the silver gel with the band of interest denoted by an arrow in Figure 1c.

3)Figure S1 panel lettering is off.

Thanks. The lettering in Figure S1 has been updated.

4)Figure 7c is not called out in the text.

We have verified that this figure is referenced in the following sentence: “Doxy.-induced osteoclastogenesis was accompanied by a ~17-fold increase in mRANKL produced by the explants (Fig. 7c).

5)The use of a period after Doxy. makes reading the manuscript difficult. I suggest using Doxy as a substitute.

Thanks, Doxy. has been replaced with Doxy throughout the MS.

6)It is unclear how many technical and biological replicates were used in the tandem mass-spec data. The table representing six samples (S1d should be b, see point 3) and the description in the figure legend is confusing. It appears the samples were taken from two biological replicates with three technical replicates each. The language should be clarified. Additional information should be added to the methods as to how the mass-spec data was filtered.

We thank the Reviewer for their critique. Please see where we have added to the legend “Data represent two biological replicates (1&2) and three technical replicates (a-c).” Moreover, please see where we have added technical information on how we performed the mass spectrometry analysis. “Bands of interest were cut from silver stained gels, destained, subjected to in-gel tryptic digest and evaluated by liquid chromatography coupled with tandem mass spectrometry (Proteomics Core, NHLBI). Mass spectrometry analysis was performed using a Orbitrap Fusion Lumos mass spectrometer (Thermo Scientific) coupled to a Dionex UltiMate 3000-nLC (Thermo Scientific) liquid chromatography system. The data were searched using Sequest within Proteome Discoverer software v1.4 (Thermo Fisher Scientific) against the UniProt human database. The results were filtered with a 1% false discovery rate at the level of proteins and peptides. Raw files were analyzed using the Mascot search engine (v2.3) with criteria: Database, Swiss-Prot (Swiss Institute of Bioinformatics); Taxonomy, eukaryote; Enzyme, trypsin; miscleavages, 2; Variable Modifications, oxidation (M), deamidation (NQ); Fixed Modifications, carbamidomethyl (C); MS peptide tolerance 10 ppm; MS/MS Tolerance, 0.02 Da. The peptide confidence false discovery rates was set to less than 1%.”

7)“La’s tight regulation during osteoclastogenesis must be carried out at the protein level, as, despite the scarcity of La protein, M-CSF derived precursors contain even more La transcript (gene SSB) than after RANKL application (Fig. S1e).”

•The authors have shown that osteoclastogenesis relies on the appropriate post-translational species and trafficking of La-protein. However, they cannot conclude that other forms of post-transcriptional regulation contribute more to these events without the appropriate controls. I suggest softening the language for this sentence.

Thanks. We have edited this sentence to: “We suggest that tight regulation of La during osteoclastogenesis is carried out at the protein level, as, despite the scarcity of La protein, M-CSF derived precursors contain even more La transcript (gene SSB) than after RANKL application (Fig. S1e).”

REVIEWER 4.

We greatly appreciate the critiques and suggestions that helped us to strengthen the manuscript.

The manuscript by Whitlock et al describes the involvement of La protein (SSB) in osteoclast fusion. Osteoclasts are essential for skeletal remodelling due to bone-resorbing properties, and derive from monocytes upon stimulation with cytokines including M-CSF and RANKL. By monitoring changes in protein levels during monocyte differentiation in vitro, the authors discover an involvement of the RNA binding protein La, which they aim to characterise in the rest of the study.

Although understanding the molecular mechanism of regulated osteoclast fusion is a very important outstanding question in the field, **the current study lacks clarity, fails to present a convincing mechanism of La-protein action and moreover suffers in part from poor experimental design and lack of experimental controls.** Thus, the present study and conclusions drawn seem rather premature and add confusion instead of offering new insights into the biology of osteoclast fusion. **Moreover, the manuscript has not been assembled with sufficient care (e.g. an antibody annotation conflict Fig 1D, and incorrect in-text references to Fig3).**

Sorry for the wrong annotation of the Ab in the Fig. 1D (it should be anti-La Abcam 75927) and incorrect reference to the figure 3, both are corrected in the revised MS.

We agree that the molecular mechanism by which La promotes osteoclast fusion remains unclear. However, as noted in our reply to the 2nd comment of Rev. 1, we do consider the discovery of the regulatory role of La in osteoclastogenesis to be a major advance in the field and finding that osteoclast fusion is regulated by cell-surface complexes of La and Anx A5 connecting La function with phosphatidylserine signaling in fusion-committed cells to be an important and unexpected mechanistic insight. Furthermore, our findings identify easily accessible cell-surface La as an intriguing target for treating bone loss diseases. Complete dissection of the mechanisms of osteoclast fusion will likely take many follow up studies. Above (reply to the 2nd comment of Rev. 1) we mention an example of myomaker, for which 9 years after the discovery of myoblast fusion dependence on myomaker (Millay et al., Nature, 2013), we still do not know the underlying mechanisms.

Major concerns

1) A more unbiased quantitative approach seems warranted to monitor proteomic changes during osteoclast differentiation/fusion.

While we agree that unbiased quantitative study of proteomic changes during osteoclast differentiation/fusion will be useful for the field, this is not the goal of this work. We have identified La as a player at the early stage of our analysis and concentrated all our efforts and resources on it.

2) Given the central role of different molecular species of La protein, which the authors claim to detect by using “specific” antibodies, the antibodies in question must be validated more carefully. The fact that La protein exhibits a different MW on blots presented in figures 1 and 2 (inconsistent use of MW standards) is very confusing. Furthermore, it is unclear if the blot in 1d is for La protein or rather RANK (Abcam 13918) as stated in the figure legend. Figure 1d and 1e are supposed to be blotted for La with the same antibody, those should be run on the same gel/blot in order to give a clear idea of steady-state FL and cleaved La species during the course of osteoclast formation. In order to specifically detect FL La, the authors use a phospho-specific antibody (abcam 61800), but fail to provide evidence (on a WB) that this indeed detects specifically the full-length protein. Thus all claims made in this regard seem highly questionable. More surprisingly the authors claim to detect (the cleaved form of) La at the cell surface. However, given the absence of a previously reported or experimentally addressed mechanism of La-protein secretion as well as the strikingly different cell surface staining pattern in human and mouse cells (Fig 3d-c) better specificity controls are required (e.g. does RNAi abolish/reduce staining).

Also the Antibody PA5-29763 (which is referred to as a-LMW La) is actually raised against AA 2-242 of human SSB, and so should detect both the long and short form of the protein.

We are especially grateful to this Reviewer for drawing our attention to our confusing discussion of the anti-La antibodies used in our work. We are sorry that the validation of the antibodies (Abs) used in our studies has not been clearly explained.

Validation of the antibodies. The key conclusions of the paper are supported by Western blot analysis, fluorescence microscopy, fusion and bone resorption assays using monoclonal murine α -La antibody (α -La mAb; Abcam, 75927). The specificity of this Ab (α -La mAb;

Abcam, 75927) has been validated by the following findings: (1) in new experiments suggested by the Reviewer, we found that siRNA suppression of La expression lowers La content using Western blot analysis and immunofluorescence microscopy (Fig. S4b-d). (2) a drastic increase in La content between monocytes treated with M-CSF and RANKL vs. those treated only with M-CSF detected by mass spectroscopy analysis results in drastic increase in La bands in Westerns and a decrease in La staining (Fig. S1 and Fig. 1). (3) La immunoprecipitated with this antibody is recognized by another commercially available anti-La antibody raised against a different region of La (Fig. 5a). (4) while α -La mAb shows us cytosolic and cell surface staining in osteoclasts at the time of fusion, the same antibody gives expected nuclear staining in HeLa cells (Fig. S1f). We have included a specific discussion of the validation of this antibody into the revised MS.

This Ab has been used in the key experiments of this study showing the decrease in La levels in monocytes after treating them with M-CSF but not RANKL and reappearance of La bands of different molecular weight at the time of fusion (Fig. 1), appearance of La at the surface of fusing osteoclasts ((Fig. 3). This Ab was found to inhibit osteoclast fusion for osteoclasts formed by application of recombinant RANKL; and by co-incubation with osteoblasts; and in ex vivo model of fibrous dysplasia (Fig. 3, 7 and 8). This Ab has been also used in our experiments revealing La-Anx A5 interactions (Fig. 5) and in bone resorption experiments (Fig. 6).

In a few experiments we have used two other anti-La antibodies: rabbit polyclonal Abcam 61800 antibody that specifically recognizes phosphorylated human SSB/La (phosphoSer366) and rabbit polyclonal ThermoFisher antibody PA5-29763 raised against AA 2-242 of human SSB/La. The comments of this Reviewer have convinced us that our discussion of these Abs and the ways we named them (FL La Ab and a-LMW La Ab) were confusing and misleading. Our findings (mostly correlations between appearance and disappearance of different species of La in cell lysates and nuclear vs cytosolic localization) and the literature (the dependence of La cleavage on its dephosphorylation) supported but did not prove that Abcam 61800 preferentially recognizes FL-La and PA5-29763 preferentially recognizes LMW-La. As noted by the Reviewer PA5-29763 should detect both the long and short form of the protein. Our finding that PA5-29763 preferentially stains cytosolic and dephosphorylated La likely reflects modification of the cleaved protein. This empirically observed distinction between preferred specificities of Abcam 61800 and PA5-29763 to FL-La and LMW-La, respectively, is not important for our work. Changes in the levels and species of La and La localization from cytoplasm/cell surface to nucleus during osteoclast differentiation have been documented with α -La mAb; Abcam, 75927 that recognizes all species of La studied. In revision, we have renamed Abcam 61800 and PA5-29763 to avoid placing the suggested interpretation into the name of the reagents. Rabbit Ab Abcam 61800 specific for La phosphorylated at Ser366 is now referred to as α -p366 La rAb (instead of α -FL La in the original version). Rabbit Ab ThermoFisher PA5-29763 is now referred to as α -La rAb (instead of α -LMW La). We have also correspondingly modified the discussion of the experiments with these Abs.

In response to the differences in molecular weight standards, we apologize. Over the last three years during this work a variety of reagents became difficult to obtain due to global supply chain issues. We initially used a molecular weight standard from Bio Rad and substituted a standard from Biodynamics that our supply center was able to obtain so that we could move forward with our work. While the standards are roughly equivalent, each differs by 5-8 KDa from Biorad.

In response to the differences in molecular weight between Figure 1 and Figure 2, all Westerns from Figure 1 represent samples from primary human cells while the Western in Figure 2 represents samples from an immortalized murine cell line. We find that human vs mouse La does exhibit a slight molecular weight difference. This empirical observation may be explained by mouse La containing an additional 7 amino acids or some difference in posttranslational modifications in human vs mouse osteoclasts. While human and mouse La share substantial sequence similarity, there are substantial differences in the amino acid sequences of the two homologues. In particular, the primary amino acid sequence of the c-terminal half of La is rather divergent when comparing the two.

3) In order to evaluate the more canonical role of RNA-binding protein La in post-transcriptional regulation of mRNAs, the authors evaluate mRNA levels of known osteoclast fusion factors. However, the literature and even the publication cited by the authors (Sommer et al 2011) mainly describes a role of La in mRNA translation. Thus, the authors must look at protein and not mRNA levels to reach a valid conclusion.

Thanks. We agree with the critique and have added Western blot analysis to characterize protein levels for Syncytin 1, DC-Stamp, Annexin A5 and TMEM16F.

4) In figure 2, the fusion promoting effect of ectopic La expression should best be evaluated in the RNAi background. This type of rescue experiment would additionally validate RNAi specificity. Additionally, the authors have to provide WB analysis of different constructs. mRNA/RT-qPCR is insufficient and likely flawed by the large excess of input/transfected DNA.

To address this concern, we have carried out experiments in which we rescued fusion inhibited by RNAi suppression of La expression by application of recombinant La* (Fig. S4e). We have now added a representative Western comparing untransfected lysates to lysates from cells transfected with La D371A, D374A, La 1-375, and La* 1-375 that demonstrates the expression of these constructs and the difference in molecular weight between D371A, D374A vs La 1-375.

5) As mentioned above the cell surface staining with anti-La presented in Figure 3 requires further validation. Furthermore, the use of abs to inhibit fusion has to be controlled better. It is not clear that the observed effect is not due to high (background) binding by anti-La, where control abs do not bind, or bind to a much lesser extent.

We appreciate the critique. We validated monoclonal α -La mAb by showing lowered cell surface labeling detected with this antibody for the cells with expression of La lowered by siRNA (Fig. S4d).

In the revised manuscript, we present new data arguing against hypothesis that anti-La Abs inhibit fusion because of their high (background) binding. As seen in Fig. 3f, α -La mAbs inhibited osteoclast fusion and anti-RANK mAbs antibodies had no effect despite comparable levels of binding between α -La mAb and α -RANK mAb to the surface of synchronized cells following LPC removal (Fig. S5a). Furthermore, α -La rabbit Abs (α -La rAb), in contrast to isotype control IgG, also block synchronized osteoclast fusion (Fig. S5b). a-p366 La Ab did not suppress fusion, suggesting that La phosphorylated at Ser366 does not contribute to the fusion stage of osteoclast formation (Fig. S5). Importantly, thanks to the strong, non-specific binding of any rabbit IgG to the abundant Fc receptors on the surface

of human macrophage-lineage cells (Ober et al., Int Immunol 13, 1551 (2001), the levels of osteoclast surface binding by all 3 rabbit antibodies: α -La rAb, a-p366 La rAb and control IgG, are similar (Fig. S5c). Thus, our finding that only α -La rAb inhibits fusion cannot be explained by non-specific steric hindrance of cell surface by associated immunoglobulins.

6) It remains unclear if the fusion stimulating effect of recombinant La (Fig.4) is exerted on the cell surface and not inside the cell. Certainly, the prolonged (overnight) incubation time with recombinant protein would allow for cellular uptake and a cytosolic effect. Thus, further control experiments are warranted.

In synchronized fusion assay, fusion-stimulating effect of recombinant La and fusion-inhibiting effect of anti-La antibody are scored 90 min after application of the protein (Fig. 4e). Note that our conclusion that osteoclast fusion is regulated by cell surface La rather than by intracellular La is supported by several lines of evidence: (i) recombinant La promotes synchronized fusion, Fig. 4e; (ii) anti-La antibodies inhibit synchronized fusion, Fig. 3f; (iii) development of fusion activity of differentiating osteoclasts correlates with an increase in cell surface La staining, Fig. 3d. We consider all these findings and our finding that recombinant full-length La and La 1-375 and La 188-375 promote fusion, La 1-187 has no effect, Fig. 4d difficult to explain within hypothesis that La-derived recombinant proteins and La-targeting antibodies influence fusion by acting inside the cells.

7) The authors have addressed whether La acts itself as a fusogen. This is described only in the text but the underlying data is not shown/has been omitted?

In the Methods we have more fully described the assay used: fibroblasts with bound, lipid probe-labeled-erythrocytes, where fusion and even early hemifusion intermediates are detected by the transfer from erythrocyte to fibroblast. In the Results, we more fully describe the data showing that none of 872 analyzed HA0-expressing fibroblasts with bound erythrocyte acquired lipid probe after application of recombinant La in 2 independent experiments. These data indicate that La does not fuse or even hemifuse cells tightly bound by fusion-incompetent hemagglutinin. In the Results, we also present the results of the statistical analysis of this data by Wilson's method, suggesting that the probability of La-mediated fusion does not exceed 0.0044 per cell contact. To address this comment of the Reviewer, we added a supplemental figure showing a representative image showing the lack of labeled fibroblasts after application of recombinant La (Fig. S7a). As a positive control, we also included an image showing the very efficient fusion (many labeled fibroblasts) mediated by fusion-competent (=trypsin-cleaved and low pH treated) influenza hemagglutinin.

We have now added new evidence challenging the hypothesis that La acts as an activated fusogen. Application of recombinant La promotes neither lipid mixing of liposomes nor fusion of myoblasts (Fig. S7b,d).

8) In order to evaluate a role for La as a fusion regulator the authors describe its interaction with Annexin A5 (a protein previously reported by the same lab to be involved in fusion). However, data presented in Fig S5 are not convincing as lysate controls are missing or not run properly alongside the immuno-precipitated samples. The data representing liposome binding experiments (FigS5 D) are confusing in terms of what has been tested but essential negative controls seem to be missing. Also, the primary data (protein gels should be shown).

More importantly however, the authors fail to provide any evidence that would help to answer the question of how the postulated interaction with Annexin A5 would stimulate osteoclast fusion.

As suggested in this comment and discussed in our response to the 2nd comment 2 of the Reviewer 1, we have deepened our analysis of the La-Anx A5 interactions. Novel and unexpected finding that La binds to Anx A5 at the surface of fusion committed osteoclasts had been supported by following lines of evidence: (1) La-containing protein complexes isolated by immunoprecipitation from osteoclast lysates contain Anx A5; (2) Anx A5-containing protein complexes contain La; (3) La binding to liposomes depends on Anx A5, Ca²⁺ and PS. Our new findings further substantiate and develop this conclusion: (4) streptavidin pulldown of recombinant 6xHis-La from mixtures of recombinant Biotin-AnxA5 and 6xHis-La; (5, 6) loss of cell surface La observed after lowering the cell surface Anx A5 by either Ca²⁺ removal or suppressing PS exposure. Our new data provide additional evidence for direct La-Anx A5 binding, explain how La, a soluble protein, associates with cell membrane, and, most importantly, connect La function in osteoclastogenesis with non-apoptotic phosphatidylserine exposure signaling pathways that regulate osteoclast fusion. We consider these findings to be an important step in unraveling the mechanisms by which signaling status of the differentiating osteoclast precursors control the timing and the size of osteoclasts. We have re-written the description of the experiments in which we explore La binding to liposomes via Anx A5 to better explain them.

In response to the Reviewer's specific comments concerning our immunoprecipitation blots. All blots shown for a particular experiment were ran side by side on the same gels/blots. In these experiments we ran a variety of input samples (e.g., with or without crosslinking) and separated them from our immunoprecipitation lanes by an additional molecular weight standard. Dashed lines simply indicate that we selected lanes of interest and placed them side-by-side.

In general, Anx A5 is an abundant protein in these cells at fusion timepoints. We expect that the majority of Anx A5 is within the cytosol of OCs executing functions that have nothing to do with La. In the immunoprecipitation experiments in Figure S6a, we omitted the input lane so as not to show some large uninterpretable black spot. The point of the Figure S6a is to illustrate that when La or Anx A5 are immunoprecipitated they bring along their binding partner (which does not happen when using isotype control antibodies). We show the input lanes from Anx A1 and Anx A4 simply to demonstrate that our antibodies raised to these proteins work, but that we do not detect these proteins when immunoprecipitating La. At this time, we are unable to comment on the fraction of La that binds Anx A5 or vice versa. We simply demonstrate that these two proteins do interact in this specific biological context and go on to demonstrate that the proteins can directly bind one another in subsequent figures.

In response to the Reviewer's specific comment on our liposome binding assays, first we apologize for not explaining the experiments clearly enough to be easily interpreted. Please see where we have amended our previous description of these experiments in our methods section. In addition, please see where we have added a representative gel showing the levels of La or Anx A5 in both soluble and pelleted fractions for the conditions described (Fig. S6b).

9) Given insufficiently controlled RNAi and anti-La surface binding/ inhibition of fusion experiments, effects observed by the authors on bone resorption (Fig 5) in a osteoblast/osteoclast

co-culture setting (Fig 6) and in a mouse model exhibiting an excess of osteoclast fusion (Fig 7), do not really add to the story, as unfortunately they cannot be interpreted with sufficient confidence.

We appreciate the Reviewer's concerns but hope that additional experiments validating and explaining our approaches have addressed these concerns. As discussed above in our response to the 2nd comment of this Reviewer, we consider the anti-La antibody used in Fig. 5, 6 and 7 (α -La mAb, Abcam, 75927) to be well validated, including by the new experiments confirming the suppression of the surface La staining by RNAi and by the fusion rescue experiments validating our RNAi approach (Fig. S4e). Furthermore, both new data and, hopefully, better explanations confirm the specificity of the fusion inhibition by anti-La antibodies. Findings in Fig. 6, 7 and 8 support our conclusion that cell surface La regulates not only cell fusion but also bone resorption, and not only osteoclast formation induced by recombinant RANKL but also osteoclast formation triggered by factors released by osteoblasts both in osteoclast/osteoblast cocultures and in an ex vivo model of fibrous dysplasia.

Reviewer #1 (Remarks to the Author):

In the revised manuscript, the authors performed additional experiments and provided more detailed mechanisms underlying how La and Anx5 may regulate osteoclast fusion.

The authors examined the effect of RNAi suppression of La on the expression levels of other fusogenic factors, and the role of La in myocyte fusion was also addressed in revision.

However, the main message of this manuscript remains unproven. Although the reviewer asked to show the physiological relevance of La in osteoclast fusion in vivo, the authors did not address this most important point.

The authors claimed that it is difficult to examine the role of La in vivo because La also functions as an RNA chaperone, but I do not think it reasonable explanation for not performing the in vivo experiments. It will be possible to generate conditional knockout mice or the mice lacking only fusion-related domain of La, which will make the paper much more convincing.

Given that the main message of this work is that La is important for osteoclast fusion but there is no sufficient evidence proving it, the revised manuscript is not suitable for publication in Nature Communications.

Reviewer #3 (Remarks to the Author):

Summary

The authors report that the spatial-temporal regulation of La protein contributes to monocyte-to-osteoclast differentiation. While La protein is typically thought to play roles in RNA metabolism, the manuscript describes a new direct or indirect contribution of La protein in osteoclast formation. Upon phosphorylation at S366 and caspase cleavage, a low molecular weight species of La is present at the cell surface. Proteomics identified that the levels of this low-molecular weight species of La correlates with multinucleated osteoclasts. This function appears to be independent of the N-terminal binding domains of La proteins. The authors demonstrate that fusion correlates with the levels of La. Moreover, the authors show an ex vivo model of osteoclast formation in fibrous dysplasia with mutations in Gas. With this model, the manuscript displays that the application of anti-La antibodies can reduce osteoclast fusion, suggesting a possible use of La protein as a potential therapeutic target. While some of the findings in the paper are mostly correlative and must be presented carefully as such, the possible implications are of interest.

The authors have addressed my major concerns regarding the manuscript.

Reviewer #5 (Remarks to the Author):

Comments:

The authors present very interesting observations suggesting a new functional role for La in osteoblast fusion by using various experimental approaches. The findings are relevant and of interest to the field. Although the quality of the manuscript probably improved after revision, some experiments and data are still lacking clarity and convincing conclusions.

The identification of La as the protein species within the gel band is great, but the information provided within the manuscript is not sufficient. Shotgun proteomic analyses of gel bands from complex lysates usually result in the identification of many proteins, and certain filter, such as protein abundance rank, peptide number (unique peptides!), coverage, enrichment over control band etc, have to be applied for solid matching and target identification. Were La and Vimentin the only proteins identified? Was La the most abundant protein? Based on the staining signal, La should be much more abundant compared to Vimentin. Is this reflected in the mass spec data?

From the current version of the manuscript it is not clear, how the mass spec data (protein lists) were analyzed/filtered after RAW data analyses. More details would be very much appreciated.

The binding of La to AnxA5 is a major result, but despite the effort to show evidences for direct La-AnxA5 interaction (Fig5a,b, d), data still lack strength. The AnxA5 Co-IP experiment with La (Figure 5a) would benefit from additional controls. Do the functionally different or similar La variants (1-187, 188-375, 1-408) bind AnxA5? Does AnxA5 binding correlate with function in osteoblast fusion? The western blot signal of La Co-IP with biotinylated AnxA5 in Fig 5b is not very convincing. A proper control (unrelated biotinylated protein, for example Anx1 or Anx4) should be included in order to show specificity. The La western blot signal in Figure S6a (AnxA5 pulldown) is not very convincing, neither. Data presentation should be improved. Is the signal with IgG control background or La specific?

The liposome assay (Figure 5c, S6b) seems to be an elegant assay, but lacks controls (for example nonfunctional La 1-188 construct, other annexins) in order to be completely convincing.

The Anx5 depletion essay (Fig 5g) as an approach to confirm La interaction and binding on the cell surface is rather indirect and lacks experimental design. EGTA and A01 treatment (Fig5e,f) leads to decrease in AnxA5 and La cell surface signal. The AnxA5 dependency of La surface binding is not shown. La itself could be Calcium and PS dependent. The experiment should be expanded, adding recombinant La protein, even including non-functional La constructs (1-188), to EGTA/A01 treated (AnxA5 depleted) cells.

The lack of La in mediating myoblast cell fusion (Fig S7) is a quite relevant and interesting observation, and actually provides tools to further investigate the function of La in cell fusion. How do the authors explain this observation? Is La expressed in myoblasts, but not surface localized? Does recombinant La actually binds to AnxA5 and/or to myoblast cell surface? As far as I understand the figures, this was not shown. These questions can be easily addressed, for example by immunostaining and outcome is actually crucial for interpretation of data presented in this study (results of anti La mAB and recombinant La).

Assuming La and Anx A5 are expressed in myoblasts, they seem not to be enough to drive cell fusion and additional factors are required. Here an unbiased proteomic approach, such as affinity pulldown combined with mass spectrometry, would allow characterization of La interactomes of functionally different La variants as well as in different cellular background.

The authors should be careful with the interpretation of a-p366 La antibody results (Figure S5b,c). This antibody binds to La when phosphorylated at position 366. According to statements within this study phosphorylated La exists as full length protein and exclusively localizes to the nucleus, therefore it is not cell surface exposed. Although the authors state on page 10 line 2, that antibody binding was similar, including anti p366 La antibody, the data for that phosphorylation specific antibody are not shown (surface staining), and to my understanding phosphorylated La is not supposed to localize at the membrane.

First, I wonder what the rationale was for using the anti p-366 La antibody in this essay? Using the anti La mAB as an additional antibody would have made more sense, since at least it recognizes the surface exposed La variant.

Second, interpretations should be done carefully. Making conclusion about phosphorylation status is not appropriate and rather indirect, since phosphorylated La was not shown to be surface localized so far, and impact of phosphorylation (for example using cleavage mutant with phosphomimetic mutation) was not investigated.

Specific comments:

Images and western blots for La in murine cells (Figure 2a/b) show no nuclear localization of La as well as only one La species at day 3, in contrast to human cells, where an higher molecular weight La species is accumulated in the nucleus at later time points. How do the authors explain that difference?

In Figure S3a, western blot signal of La in untreated cells shows strong full length La signal, but the corresponding images do not show any nuclear signal, which is inconsistent with data from Figure 1g, where at day 3 nuclear signal of La was observed. How can this be explained?

Minor:

When addressing La phosphorylation, the authors should call it non-phosphorylated La, since dephosphorylation implies a phosphorylation event before that, which might not be the case here. It rather seems to be a La post-translational modification happening in the nucleus.

Reviewer #6 (Remarks to the Author):

In the revised manuscript, the authors have addressed most of the questions by Reviewer #4. There are only a few minor points that need to be clarified:

1. The size of La in human osteoclasts.

- In Figure 1c, the band used for mass spec was above 50 kDa and the identified protein was La. However, in Figures 1e, both FL and LMW La proteins were below 50 kDa.
- In Figure 1d, two bands were in the monocyte lysate lane. Were they FL and LMW La proteins, or the FL La and a non-specific upper band? If the former, did the author detect significant cytoplasmic staining of La in the monocyte? If the latter, could the authors provide some explanation on the absence of the non-specific band in the other two lanes, as well as the absence of LMW La in the RANKL sample?

2. SSB expression and La protein level in the M-CSF sample.

- In Figure S1e, there were ~2 fold SSB transcripts in the M-CSF than the RANKL sample, but there was no detectable La protein in the M-CSF sample (Figure 1d). Is this difference due to translation inhibition or protein degradation in the M-CSF sample?
- In Figure 2e, a D1 sample after RANKL treatment should be included.

3. Localization of FL La vs. p366 La.

- In Figure S3, there was no LMW La and strong p366 La in the nucleus after caspase inhibitor treatment. Is the FL La also only localized in the nucleus on Day 3?

4. Localization of Anx A5, La and PS.

- In Figures 5 and S6, in the absence of Ca²⁺, no Anx A5 associated with liposome but 15% of La did. Does this mean that some La can interact with PS in the absence of AnxA5?
- The immunofluorescent staining showed strong Anx A5 surface staining (Figure 5e) but weak La staining. Do these proteins co-localize?
- In Figure 5e, does the strong staining of Anx A5 suggest that the cell surface of human OCs is enriched with PS without inducing apoptosis?

POINT-BY-POINT TO THE REVIEWER COMMENTS.

We would like to thank the Editor and all Reviewers for the careful review of our manuscript and the insightful and valuable comments and critiques offered. They have greatly strengthened our manuscript.

The list of new experiments/figures added in Revision includes:

Fig. 5e,f & Fig. S6k (Anx A5 suppression experiments); Fig. S3a-c (caspase 3 inhibitor z-DEVD); Fig. S5c (addition of p366 La rAB non-specific staining) Fig. S6b-d (pull-downs with recombinant Actin, La 1-187 and La 188-375); Fig. S6h,i,j (additional experiments with protein-to-liposome binding assay); and Fig. S7c,d,e,h (additional characterization of La in myoblasts).

Please see our reply to each of the comments in bold red font.

Reviewer #3 (Remarks to the Author):

Summary

The authors report that the spatial-temporal regulation of La protein contributes to monocyte-to-osteoclast differentiation. While La protein is typically thought to play roles in RNA metabolism, the manuscript describes a new direct or indirect contribution of La protein in osteoclast formation. Upon phosphorylation at S366 and caspase cleavage, a low molecular weight species of La is present at the cell surface. Proteomics identified that the levels of this low-molecular weight species of La correlates with multinucleated osteoclasts. This function appears to be independent of the N-terminal binding domains of La proteins. The authors demonstrate that fusion correlates with the levels of La. Moreover, the authors show an ex vivo model of osteoclast formation in fibrous dysplasia with mutations in *Gas*. With this model, the manuscript displays that the application of anti-La antibodies can reduce osteoclast fusion, suggesting a possible use of La protein as a potential therapeutic target. While some of the findings in the paper are mostly correlative and must be presented carefully as such, the possible implications are of interest.

We agree that the established link between fusion efficiency and cell surface concentrations of La and Anx A5 at the surface of fusion-committed osteoclasts suggests rather than proves a causative role for these proteins in fusion and added a following statement to the discussion. “Note that the established link between fusion efficiency and cell surface concentrations of La and PS-bound Anx A5 at the surface of fusion-committed osteoclasts remains correlative and only future work uncovering the molecular mechanisms of osteoclast fusion will clarify the specific role of these proteins in this fusion pathway.”

The authors have addressed my major concerns regarding the manuscript.

Reviewer #5 (Remarks to the Author):

Comments:

The authors present very interesting observations suggesting a new functional role for La in osteoblast fusion by using various experimental approaches. The findings are relevant and of interest to the field. Although the quality of the manuscript probably improved after revision, some experiments and data are still lacking clarity and convincing conclusions.

The identification of La as the protein species within the gel band is great, but the information provided within the manuscript is not sufficient. Shotgun proteomic analyses of gel bands from complex lysates usually result in the identification of many proteins, and certain filter, such as protein abundance rank, peptide number (unique peptides!), coverage, enrichment over control band etc, have to be applied for solid matching and target identification. Were La and Vimentin the only proteins identified? Was La the most abundant protein? Based on the staining signal, La should be much more abundant compared to Vimentin. Is this reflected in the mass spec data? From the current version of the manuscript it is not clear, how the mass spec data (protein lists) were analyzed/filtered after RAW data analyses. More details would be very much appreciated.

We thank the Reviewer for drawing attention to the inadequate description of our identification of La from bulk osteoclast lysates. To address Reviewer's critique, we have added a special subsection "mass spectrometry" to the Biochemistry section of our Methods with a detailed description of our analysis of the mass spectrometry data. Specifically, this section details how we went from a list of 113 identified proteins to a specific candidate to investigate. Moreover, we have now added Supplementary Table 1, which includes the list of hits identified in each sample and the enrichment, peptide number, unique peptides, and other quantitative information that the Reviewer has requested. We believe the expanded methodological description and the Supplementary Table will allow a reader to see how we parsed our hits.

The binding of La to AnxA5 is a major result, but despite the effort to show evidences for direct La-AnxA5 interaction (Fig5a,b, d), data still lack strength. The AnxA5 Co-IP experiment with La (Figure 5a) would benefit from additional controls. Do the functionally different or similar La variants (1-187, 188-375, 1-408) bind AnxA5? Does AnxA5 binding correlate with function in osteoblast fusion? The western blot signal of La Co-IP with biotinylated AnxA5 in Fig 5b is not very convincing. A proper control (unrelated biotinylated protein, for example Anx1 or Anx4) should be included in order to show specificity. The La western blot signal in Figure S6a (AnxA5 pulldown) is not very convincing, neither. Data presentation should be improved. Is the signal with IgG control background or La specific?

We thank the Reviewer for these suggestions. To the Reviewer's first point, we have now added additional La+Anx A5 pulldown experiments to test whether La 1-187 and La 188-375 bind Anx A5 (Fig. S6c,d). We find that both halves of La 1-375 have some affinity for Anx A5, suggesting that the La-Anx A5 binding site includes residues in both halves, or, alternatively, that La has more than one Anx A5 binding site. In the future, we plan to use more quantitative approaches to determine the binding affinity of La/La domains and Anx A5 that will be more informative than the current qualitative pulldown assays. These combined with mutational approaches will allow us to fully resolve the La-Anx A5 binding sites, however, we consider further identification and characterization of the binding sites in both La and Anx A5 to be outside the scope of our current story.

To the Reviewer's second point, we agree that a better control to show the specificity of the La-Anx A5 interaction is needed. In our revision, we have repeated these experiments with biotinylated actin (Figure S6b) and found that La does not bind this unrelated protein. We very much liked the Reviewer's suggestion to try other annexins, including Anx1 or Anx4. Unfortunately, the only reagent we could find was biotinylated Anx A1 from Bon-Opus (or 3rd parties that resale this same reagent). We ordered this product immediately after receiving the Reviewer's suggestion, but it is back ordered, and we still do not have an estimated delivery date.

To the Reviewer's third point, in hindsight, we agree with the Reviewer's assessment of this immunoprecipitation (original Fig. S6a). Since immunoprecipitation of native proteins is a rather challenging undertaking that requires excellent antibodies, immunoprecipitation studies most often use overexpression of FLAG-tagged proteins for these approaches. Ideally, one needs an antibody raised in one species that has very high affinity for a native protein and antibodies raised in another species that has good affinity for the denatured protein. Unfortunately, we do not have such a set of antibodies for Anx A5 and La. If antibodies are from the same species, one runs into problems with recognizing the denatured heavy and light chains of the antibody used for the immunoprecipitation while western blotting. To make matters worse, La is close in size to IgG heavy chain, and Anx A5 is close in size to IgG light chain. We have a mouse Anx A5 antibody that works well for immunoprecipitation, but not for blotting. When we immunoprecipitated Anx A5 with this Ab we need to blot with our mouse La antibody (α -La mAb). When we did this, we also get a second higher band in both isotype control and Anx A5 immunoprecipitation lanes that is IgG heavy chain. We believe that is what is seen in the IgG lane you referenced. We have spent considerable time trying to immunoprecipitate Anx A5 with a rabbit antibody. We very nicely see that, in contrast to isotype control, this Anx A5 rabbit antibody also pulls down La. However, now we need to use our mouse Anx A5 antibody to confirm that Anx A5 is immunoprecipitated, and have found that it does not recognize denatured Anx A5 well in our hands. Due to all these technical issues and the low quality of the data from these immunoprecipitations, we have removed the Anx A5 immunoprecipitation from Figure S6. We believe our La immunoprecipitations (Fig. 5a, Fig. S6a), Anx A5 pulldowns (Fig. 5b, Fig. S6b), Anx A5 liposome enrichments (Fig. 5c,d, Fig. S6e-j), EGTA (Fig. 5g,h), A01 (Fig. 5i) experiments and new Anx A5 knockdown data (Fig. 5e,f, Fig. S6k) strongly support the interaction between La and Anx A5 in fusing osteoclasts.

The liposome assay (Figure 5c, S6b) seems to be an elegant assay, but lacks controls (for example nonfunctional La 1-188 construct, other annexins) in order to be completely convincing.

Thank you. As suggested, we improved the controls used for our liposome sedimentation assay. To further demonstrate that the ability of AnxA5 to enrich La on PS containing membranes is specific, we used Actin as a negative control. We found that the addition of Actin does not enrich La on PS containing liposomes (Fig. S6j). We did not try recombinant portions of La (La 1-187 and La 188-375) in liposome experiments because we have explored these recombinant proteins in pull-down experiments (see above) and found both to have some affinity for Anx A5. We agree that other annexins would be a very nice comparison to explore the specificity of the Anx A5-La interactions, however, we were

unable to obtain recombinant versions of these proteins in a timely manner, as discussed above.

The Anx5 depletion essay (Fig 5g) as an approach to confirm La interaction and binding on the cell surface is rather indirect and lacks experimental design. EGTA and A01 treatment (Fig5e,f) leads to decrease in AnxA5 and La cell surface signal. The AnxA5 dependency of La surface binding is not shown. La itself could be Calcium and PS dependent. The experiment should be expanded, adding recombinant La protein, even including non-functional La constructs (1-188), to EGTA/A01 treated (AnxA5 depleted) cells.

As suggested by the Reviewer, we have strengthened the analysis of the Anx A5 dependence of La-to-cell surface binding by specifically targeting Anx A5 expression in primary osteoclasts with siRNA. We knocked down Anx A5 expression in fusion-committed osteoclasts and found that loss of Anx A5 dramatically reduced cell surface La and osteoclast fusion (Fig. 5e,f). We confirmed that these effects were not a result of gross alterations in the steady-state level of La transcript when we reduced Anx A5 transcript (Fig. S6k). Now the Anx A5 dependence of surface La in differentiating osteoclasts is supported by three independent and complementary approaches. We consider the suppression of Anx A5 expression approach to be more straight-forward than the more complex and difficult to interpret experiments adding recombinant La protein to EGTA/A01 treated cells. These suggested experiments are very interesting, but unfortunately come with technical challenges. Specifically, both EGTA and A01 treatments impact the attachment of the primary osteoclasts to coverslips. Hence, the additional time required to incubate the cells with recombinant protein, carry out washes and then fix the cells causes us to lose too many cells. Our new data on the AnxA5 dependency of surface La on osteoclasts considerably strengthens the conclusions based on pull-down, immunoprecipitation, EGTA and A01 experiments in the earlier version of this manuscript.

The lack of La in mediating myoblast cell fusion (Fig S7) is a quite relevant and interesting observation, and actually provides tools to further investigate the function of La in cell fusion. How do the authors explain this observation? Is La expressed in myoblasts, but not surface localized? Does recombinant La actually binds to AnxA5 and/or to myoblast cell surface? As far as I understand the figures, this was not shown. These questions can be easily addressed, for example by immunostaining and outcome is actually crucial for interpretation of data presented in this study (results of anti La mAB and recombinant La).

Assuming La and Anx A5 are expressed in myoblasts, they seem not to be enough to drive cell fusion and additional factors are required.

As suggested by the Reviewer, we have deepened the analysis of La in myoblast fusion. Addressing the Reviewers second question, we found C2C12 cells to express La at similar levels in proliferating and differentiating cells (Fig. S7c). We immunostained permeabilized, differentiating myoblasts for La and found it to be distributed throughout these cells, in contrast to the strong concentration of La at the rim of differentiating Raw 264.7 osteoclasts (Fig. S7d). In experiments on non-permeabilized cells, we found some La at the surface of differentiated C2C12 cells but the amount of cell surface La for myoblasts and the ratio of fluorescence intensities for surface La (non-permeabilized cells) vs cytosolic

La (permeabilized cells) were considerably lower for C2C12 cells than for Raw 264.7 cells (Fig. S7d,e). Addressing the Reviewer's third question, we fluorescently labeled recombinant La and found C2C12 cells to show considerably lower ability to bind exogenously applied recombinant La protein than Raw 264.7 osteoclasts (Fig. S7h). These findings, along with our earlier data showing that neither α -La mAb nor recombinant La significantly affect myoblast fusion (Fig. S7d,e), suggest that, in contrast to osteoclast fusion, myoblast fusion does not depend on La externalization.

In response to the Reviewer's first question, we interpret these data as follows. Since myogenic differentiation boosts surface concentration of Anx A5 (Ref. 70), Anx A5 at the surface of myoblasts seems to be unavailable for La binding, substantiating the hypothesis that Anxs at the surface of myoblasts are bound to some, yet unidentified, regulator of myoblast fusion connecting the conserved regulatory mechanisms underlying PS signaling in fusion-committed cells with myoblast-specific components of the fusion machines. As the Reviewer puts it, "Assuming La and Anx A5 are expressed in myoblasts, they seem not to be enough to drive cell fusion and additional factors are required." In revision we added a comment on this: "Low efficiency of binding of exogenously added La to Anx A5 at the surface of myoblasts may indicate that these Anx A5 molecules are bound to another, yet-unidentified regulator of myoblast fusion connecting the conserved regulatory mechanisms underlying PS signaling with myoblast-specific components of the fusion machines."

Here an unbiased proteomic approach, such as affinity pulldown combined with mass spectrometry, would allow characterization of La interactomes of functionally different La variants as well as in different cellular background.

Agreed, but this would be a major project that we respectfully consider to be outside of the scope of this study.

The authors should be careful with the interpretation of a-p366 La antibody results (Figure S5b,c). This antibody binds to La when phosphorylated at position 366. According to statements within this study phosphorylated La exists as full length protein and exclusively localizes to the nucleus, therefore it is not cell surface exposed. Although the authors state on page 10 lane 2, that antibody binding was similar, including anti p366 La antibody, the data for that phosphorylation specific antibody are not shown (surface staining), and to my understanding phosphorylated La is not supposed to localize at the membrane.

First, I wonder what the rationale was for using the anti p-366 La antibody in this essay? Using the anti La mAb as an additional antibody would have made more sense, since at least it recognizes the surface exposed La variant.

Second, interpretations should be done carefully. Making conclusion about phosphorylation status is not appropriate and rather indirect, since phosphorylated La was not shown to be surface localized so far, and impact of phosphorylation (for example using cleavage mutant with phosphomimetic mutation) was not investigated.

We very much appreciate these critiques and agree that our description of these data was unclear. We do not expect any p366-La at the surface of these cells and use the rabbit antibody to p366-La (α -p366 La rAb) as a negative control, additional to the non-specific

IgG. All rabbit antibodies used: α -La rAb, α -p366 La rAb (we added this image to Fig. S5c in this revision) and rabbit IgG, similarly stain the surface of the cells because of the strong, non-specific binding of any rabbit IgG to the abundant Fc receptors on the surface of human macrophage-lineage cells (Ref. 47). Despite the similar binding, α -La rAb inhibits fusion and neither α -p366 La rAb nor control IgG does. Thus, our finding that only α -La rAb inhibits fusion cannot be explained by non-specific steric hindrance of cell surface associated immunoglobulins. Furthermore, even if p366 La is present at the surface of the cells (we do not expect this but cannot exclude it because of the non-specific staining by rabbit antibodies), it does not contribute to fusion. We have re-written the description of these data in the revised MS to better explain them.

Specific comments:

Images and western blots for La in murine cells (Figure 2a/b) show no nuclear localization of La as well as only one La species at day 3, in contrast to human cells, where an higher molecular weight La species is accumulated in the nucleus at later time points. How do the authors explain that difference?

As we discuss in the manuscript, murine La is important for murine osteoclast fusion and some of the regulatory changes in La, including its cytosolic localization; a drastic increase in its steady-state levels; significant La surface staining at the time of fusion; as well as fusion promotion after overexpression of La 1-375, are conserved between human and murine cells. However, there are also some differences between what we observe with La in human vs murine osteoclasts. We see in Figure 2b, that in contrast to human cells, in murine cells there is a consistent, low-level expression of a higher molecular weight La at all timepoints. Moreover, in murine cells, we do not see a stark shift from LMW La to FL La between days 3 and 5. Instead, we see an upregulation of LMW La at D3 followed by a reduction of LMW La level by D5. Furthermore, we do not see a general accumulation of La in the nuclei of murine osteoclasts at D5, but we do see La in their nuclei as distinct speckles within nuclei (La has been reported to have a nucleolar localization signal, so it may be in nucleoli) in Figure 2a. These differences between our findings in human and murine osteoclasts can be explained by several reasons. To start with, we use primary human cells and immortalized (RAW 264.7) murine cells. The shift in fusion plateauing in primary human cells at later timepoints is much cleaner than in these immortalized cells that still maintain many mononucleated, non-fusogenic cells at all timepoints (a well-known feature of these cells). We believe some of the differences we see may be because one model better recapitulates the regulatory shifts between precursor, fusogenic cells and mature, multinucleated osteoclasts where fusion has reached a plateau. It is also possible that some of the differences between human and murine osteoclasts can be caused by the fact that throughout our study we have used antibodies raised to human La. While murine and human Las share high sequence similarity, there are significant differences at the primary amino acid sequence that may result in our not seeing the whole picture in murine osteoclasts. This is a perineal shortcoming in the La community, as generation of good antibodies for murine La have, for some unexplained reason, been much more challenging and such antibodies are not as readily available for purchase. Finally, it is possible that murine and human osteoclasts have evolved to regulate the function of La in osteoclast

fusion in subtly different ways. Seeing that at later fusion timepoints La does return to the nucleus of murine cells, but in some distinct region/speckle is very interesting and different from the general nuclear staining we see in human cells (Fig. 2a vs 1g). However, we expect that we will need to develop good antibodies to murine La to have full confidence in anything we would like to say about human vs murine La in osteoclasts. In the Revision, we now mention these subtle differences between human primary osteoclasts and murine RAW 264.7 cells and their possible explanation in the Discussion section.

In Figure S3a, western blot signal of La in untreated cells shows strong full length La signal, but the corresponding images do not show any nuclear signal, which is inconsistent with data from Figure 1g, where at day 3 nuclear signal of La was observed. How can this be explained?

We thank the Reviewer for drawing our attention to this issue. In this figure (original version of Fig. S3), our Western blot panel (Fig. S3a) and immunofluorescence image (Fig. S3b) have been prepared using different antibodies (α -La mAb and α -p366 La rAb, respectively). We noted this in the labels for both panels, but now we recognize that this has been confusing. α -La mAb recognizes both FL La and LMW La, whereas α -p366 La rAb only detects La phosphorylated at Ser366 that corresponds to FL, nuclear La. In the revision we have replaced the immunofluorescence image with new Figure S3c, using the same antibody (α -La mAb) as Fig. S3a. The new figure shows both nuclear and cytoplasmic staining and shows the shift to predominately nuclear staining following z-VAD treatment.

Minor:

When addressing La phosphorylation, the authors should call it non-phosphorylated La, since dephosphorylation implies a phosphorylation event before that, which might not be the case here. It rather seems to be a La post-translational modification happening in the nucleus.

Thanks. Corrected.

Reviewer #6 (Remarks to the Author):

In the revised manuscript, the authors have addressed most of the questions by Reviewer #4. There are only a few minor points that need to be clarified:

1. The size of La in human osteoclasts.

- In Figure 1c, the band used for mass spec was above 50 kDa and the identified protein was La. However, in Figures 1e, both FL and LMW La proteins were below 50 kDa.

- In Figure 1d, two bands were in the monocyte lysate lane. Were they FL and LMW La proteins, or the FL La and a non-specific upper band? If the former, did the author detect significant cytoplasmic staining of La in the monocyte? If the latter, could the authors provide some explanation on the absence of the non-specific band in the other two lanes, as well as the absence of LMW La in the RANKL sample?

FL La is predicted to be ~47 kDa but is known to be heavily posttranslationally modified in most cells. In our experiments, La typically runs around 48-55 kDa on Tris-glycine gels in our hands. For Fig. 1c, as stated in the legend, we separated the lysate on a Bis-Tris PAGE gel (the rest of the gels in the manuscript are Tris-glycine). We find that in Bis-Tris PAGE gels La runs at a higher molecular weight, which allowed us to better separate the band in our silver gels that we cut to submit for mass spectrometry analysis. Moreover, as stated in the legend, we ran this gel until the 50 kDa marker nearly ran off the gel front to achieve better separation, resulting in La appearing heavier than it actually is.

As for the monocytes, yes, we routinely see both FL and LMW La in cell lysates. We would like to emphasize that we receive these primary, elutriated monocytes from a core facility here at NIH, as described, and that they are not sorted for live vs apoptotic cells. For us, this is not a problem, because the apoptotic cells do not attached to the dish when we add M-CSF. However, because we have shown caspase 3 is critical for the production of LMW La, if there exists an elevated level of apoptotic cells in these monocyte samples (likely), then they will probably cleave La as they apoptose as has been observed previously (Ref. 33). We were unable to evaluate intracellular distribution of La in these cells because they are non-adherent, and therefore incompatible with the staining protocols we use. The role of FL vs LMW La in monocytes and macrophage precursors is very interesting. We hypothesize that it is likely doing something important for the differentiation of these cells into the osteoclast lineage. However, we chose to focus on the fusion stage of osteoclast formation in this manuscript and view these intriguing questions as outside the scope of the current manuscript.

2. SSB expression and La protein level in the M-CSF sample.

- In Figure S1e, there were ~2 fold SSB transcripts in the M-CSF than the RANKL sample, but there was no detectable La protein in the M-CSF sample (Figure 1d). Is this difference due to translation inhibition or protein degradation in the M-CSF sample?

We have not done the appropriate experiments to examine the mechanism of how osteoclast precursors exhibit abundant transcript but negligible La protein. This is a phenomenon known for many transcripts/proteins throughout cell biology and could be the result of translational inhibition, the sequestration of these transcripts (e.g., into RNA speckles) or the rapid degradation of La protein within osteoclast precursors. At this stage we cannot confirm or exclude any of these mechanisms but, since we do not see laddering or a smear of La signal at lower molecular weights (corresponding to degraded La fragments) we consider the degradation scenario less likely. We now mention these mechanisms in the revised manuscript.

If the Reviewer is concerned that the difference between the RNA vs Protein levels we observe is because La is degraded in our samples as/after we collect them, 1) We have repeated these experiments with consistent results more than a dozen times at this point. The M-CSF sample would have to be uniquely degraded in a specific fashion (in comparison to the RANKL stimulated samples we also collect) each time. 2) If the signal were lost due to degradation of La in the sample we would expect to see laddering or a smear of La signal at lower molecular weights (corresponding to degraded La fragments),

and we never have. Moreover, if La were being degraded specifically in the M-CSF sample and not the other samples this degradation would have to be specific to La, as we do not see loss, laddering or smearing of other proteins in these samples (e.g., the loading control used). 3) Finally, we also see a loss of La signal when we use the same or different antibodies to stain La in M-CSF alone treated cells via immunofluorescence imaging (Fig. 1f) Together, we believe that our data strongly support the interpretation that La is controlled at the transcript level in osteoclast precursors, but the mechanism of this regulation remains to be clarified.

- In Figure 2e, a D1 sample after RANKL treatment should be included.

We believe the Reviewer is commenting on the time course of La expression in Figure 1e. In most cases we see no La at D1. However, at this early stage of differentiation, the levels, while remaining lower than at D2, considerably vary from donor to donor. In some donors we do see a low level of LMW. Most importantly, we never observe fusion at D1 and our focus in this study is on the correlation between La levels and fusion. While the role of M-CSF-induced loss and RANKL-induced reappearance of La at the very onset of monocyte-to-osteoclast differentiation is a very interesting subject, we consider it to be outside of the scope of this paper and respectfully decided against adding D1 to the figure.

3. Localization of FL La vs. p366 La.

- In Figure S3, there was no LMW La and strong p366 La in the nucleus after caspase inhibitor treatment. Is the FL La also only localized in the nucleus on Day 3?

Unfortunately, we found no commercially available antibodies that specifically recognize FL La vs LMW La in immunofluorescence staining. We expect that like for other cell types where this question has been addressed, most of FL La is nuclear for all timepoints. Note that, as also discussed in our response to Reviewer 5, in this figure (original version of Fig. S3), Western blot panel (Fig. S3a) and immunofluorescence image (Fig. S3b) have been prepared using different antibodies (α -La mAb and α -p366 La rAb, respectively). We noted this in the labels for both panels, but now appreciate that this has been confusing. α -La mAb recognizes both FL La and LMW La, whereas α -p366 La rAb only detects La phosphorylated at Ser366 that corresponds to FL, nuclear La. In the revision, we have replaced the immunofluorescence image with a new Figure S3c, using the same antibody (α -La mAb) as Fig. S3a. The new figure shows both nuclear and cytoplasmic staining and shows the shift to predominately nuclear staining following z-VAD treatment. We have also added experiments with z-DEVD, a specific inhibitor of caspase 3. Finding that this inhibitor, similarly to z-VAD (pan-caspase inhibitor), suppresses the production of LMW La (Fig. S3b), substantiating our hypothesis that cleavage of La involves caspase 3.

4. Localization of Anx A5, La and PS.

- In Figures 5 and S6, in the absence of Ca²⁺, no Anx A5 associated with liposome but 15% of La did. Does this mean that some La can interact with PS in the absence of AnxA5?

Thanks. Indeed, while our data show no binding of La to PS-containing liposomes in the absence of Anx A5 (Fig. S6b), in the presence of Anx A5 but without Ca²⁺ we see a relatively small (~15%) amount of sedimented La but no sedimented Anx A5. We do not have any convincing interpretation for this intriguing finding. Our pull-down experiments document Anx A5-La interactions in the absence of membranes, and La, under oxidizing conditions, can form aggregates that sediment even without centrifugation (Berndt et al., Int J Mol Sci 2021, 22(7):3377). Perhaps, Ca²⁺ independent La-Anx A5 interactions nucleate aggregation of La into pelletable aggregates. Since we are afraid to overinterpret this finding and prefer not to offer unsubstantiated interpretations for it in the text, we have chosen to replace the phrase “La membrane association required Anx A5, Ca²⁺ and PS “ with “La membrane association depended on Anx A5, Ca²⁺ and PS”.

- The immunofluorescent staining showed strong Anx A5 surface staining (Figure 5e) but weak La staining. Do these proteins co-localize?

Yes, they do, however not everywhere. There is some fine choreography in where, when and for how long Anx A5 and La co-localize at the osteoclast membrane during fusion. Unfortunately, this has been a very challenging aspect of our system to resolve because surface staining in our primary osteoclast system is not compatible with rabbit antibodies due to their non-specific binding to the abundant Fc receptors on the surface of human macrophage-lineage cells (Ref. 47). Because both antibodies we use for surface staining in these cells are murine we have not been able to use them at the same time with traditional secondary antibodies. We are currently exploring new antibodies and direct labeling techniques for our previous ones, but it will take us considerable time to adequately evaluate the specificities of our new reagents and fully understand this relationship.

- In Figure 5e, does the strong staining of Anx A5 suggest that the cell surface of human OCs is enriched with PS without inducing apoptosis?

Yes, differentiation and fusion of osteoclast precursors is accompanied by and depends on the non-apoptotic exposure of PS, as reported in Ref. 18, showing that this PS exposure correlates with neither activation of caspases 3 or 7 nor with cell nuclei labeling by the membrane-impermeable probe TO-PRO-3, both characteristic features of apoptotic cells.

Reviewer #5 (Remarks to the Author):

The authors addressed my concerns in a satisfactory way. I support the publication of this revised manuscript

Reviewer #6 (Remarks to the Author):

The authors have addressed most of my concerns in the revised manuscript. Regarding point #4, to show the co-localization of Anx A5 and La, the authors could use isotype-specific secondary antibodies, since the primary antibody used for AnxA5 is IgG1 and for La is IgG2a. Co-staining using isotype-specific secondary antibodies should allow the authors to distinguish the AnxA5 and La signals.

Point-by-point response to the Reviewer 6 comments

Please see our reply in bold red font.

Reviewer #6: The authors have addressed most of my concerns in the revised manuscript. Regarding point #4, to show the co-localization of Anx A5 and La, the authors could use isotype-specific secondary antibodies, since the primary antibody used for AnxA5 is IgG1 and for La is IgG2a. Co-staining using isotype-specific secondary antibodies should allow the authors to distinguish the AnxA5 and La signals.

Thank you very much for your advice to use isotype-specific secondary antibodies to differentially label Annexin A5 and La. We will try this approach along with an approach of direct labeling of the primary antibodies that we use now in our project on characterization of the membrane domains enriched in La and Anx A5. We consider this characterization as a separate project, in which we also explore the effects of different La-targeting treatments. We have preliminary observations (not yet of publication quality) confirming co-localization of La and Anx A5 in the same plasma membrane patches. However, since we do have evidence for direct molecular interactions between La and Annexin A5 (pull-down and co-immunoprecipitation experiments, and several additional experimental approaches), we do not expect any indications of potential interactions that we can get from routine microscopy-based co-localization analysis to significantly strengthen the case for La-Anx A5 interaction.